# Dynamic full-field optical coherence tomography module adapted to commercial microscopes allows longitudinal in vitro cell culture study

Tual Monfort [1,2,3], Salvatore Azzollini [1], Jérémy Brogard[1], Marilou Clémençon [1], Amélie Slembrouck-Brec[1], Valerie Forster[1], Serge Picaud [1], Olivier Goureau [1], Sacha Reichman [1], Olivier Thouvenin[4,5] & Kate Grieve [1,2,3,5✉]

Dynamic full-field optical coherence tomography (D-FFOCT) has recently emerged as a label-free imaging tool, capable of resolving cell types and organelles within 3D live samples, whilst monitoring their activity at tens of milliseconds resolution. Here, a D-FFOCT module design is presented which can be coupled to a commercial microscope with a stage top incubator, allowing non-invasive label-free longitudinal imaging over periods of minutes to weeks on the same sample. Long term volumetric imaging on human induced pluripotent stem cell-derived retinal organoids is demonstrated, highlighting tissue and cell organization processes such as rosette formation and mitosis as well as cell shape and motility. Imaging on retinal explants highlights single 3D cone and rod structures. An optimal workflow for data acquisition, postprocessing and saving is demonstrated, resulting in a time gain factor of 10 compared to prior state of the art. Finally, a method to increase D-FFOCT signal-to-noise ratio is demonstrated, allowing rapid organoid screening.

[1] Sorbonne Université, INSERM, CNRS, Institut de la Vision, 17 rue Moreau, F-75012 Paris, France. [2] CHNO des Quinze-Vingts, INSERM-DGOS CIC 1423, 28 rue de Charenton, F-75012 Paris, France. [3] Paris Eye Imaging Group, Quinze-Vingts National Eye Hospital, INSERM-DGOS, CIC 1423, 28 rue de Charenton, Paris 75012, France. [4] Institut Langevin, ESPCI Paris, Université PSL, CNRS, 75005 Paris, France. [5] These authors contributed equally: Olivier Thouvenin, Kate Grieve. ✉email: kate.grieve@inserm.fr

The development of microscopy methods dedicated to cell cultures and explants has transformed our understanding of human biology[1–5]. Imaging cells and tissues outside the body remove the constraints of in vivo imaging, enabling higher spatial resolution and the use of exogenous markers[6,7], thereby permitting imaging of structures ranging from cells and sub-cellular organelles[8–14] to single-proteins[15,16]. If the high-resolution imaging of the three-dimensional (3D) structural organization of life has brought valuable information[4,15], the more recent quantification of the temporal dynamics of cells and tissues, made accessible by live cell imaging, gives functional information and a more physiological view of biological behaviors[17–23]. Many processes, including cell division rate[21], proliferation, motility, migration, differentiation, or cell death[24], can be used as biomarkers of the physiology of cells and tissues[22–24]. Deregulations of these processes are often associated with diseases such as cancer[25–27], autoimmune disorders[28,29], neurological disease[30,31], and chronic inflammation[32]. Live cell imaging of the functional cell response to physical and chemical stimuli can help to understand pathological mechanisms and evaluate response to treatments[33] or stimuli[34,35], and therapeutic efficacy[33,36].

However, for successful live cell imaging experiments, cells and tissues must be studied in a context close to their native environment, i.e., as close as possible to in vivo conditions, in order to obtain meaningful data[37–39]. Temperature, $CO_2$, $O_2$, and $N_2$ levels, and tissue culture medium composition are thus critically important[21,37–39]. Furthermore, adherent two-dimensional (2D) cell cultures are not a satisfactory model for this paradigm since the mechanical stiffness of the glass slide supporting the cells exceeds by 6 orders of magnitude the stiffness of most tissues[40,41], which has important consequences on cell organization[42], signaling[43], and fate[44]. Additionally, 2D cultures do not replicate the physiologically important long-range (>100 μm) chemical gradients and long-range 3D interactions, which play a role, for instance, in organogenesis as well as in the tumor microenvironment[25]. Explants are not completely satisfactory either, due to the difficulty in obtaining them, especially for human explants, as well as the difficulties in maintaining the explant alive once blood circulation has been sectioned, as the explants are then oxygen and nutrient-starved, leading to cell death after a few hours[45]. During the last decade, the development of differentiation protocols for growing organoids from human stem cells has been a revolution in human biology modeling[3–5,46–48]. Organoids are self-organized and self-sustaining 3D cell structures derived from embryonic stem cells or induced pluripotent stem cells (iPSCs), mimicking the 3D cellular organization and composition of the primary tissues, comprising all major cell lineages in proportions similar to those in living tissue. They replicate biologically relevant intercellular phenomena known in organs and restore some of the physiological mechanical parameters and long-range chemical and biological interactions[3–5]. Besides, patient-derived samples have become accessible in large quantities, offering outstanding opportunities for (rare) disease modeling, drug testing and development, and personalized medicine[3–5]. Despite recent advances in microscopy, high-resolution imaging of organoids, and in particular longitudinal, live, volumetric imaging, is still complex and remains an open challenge[49].

For organoid imaging, and more generally for live cell imaging, fluorescence microscopy has largely prevailed over other imaging methods[50,51]. Owing to the astounding number of fluorescent probes available, which are able to specifically label most biological entities, the spatio-temporal dynamics of virtually any structure of interest can be studied[52]. However, the use of exogenous fluorophores can often skew native cell functioning, and

they are intrinsically not suitable for live cell imaging[21,51,53–62]. These methods introduce significant artifacts, including phototoxicity, increased DNA replication stress and mitotic defects[21,51], molecular buffering[63], and displaced physicochemical equilibrium associated with preventing the formation of liquid organelles, for instance[64]. The use of genetically targeted fluorophores for live imaging is limited to a few model organisms in which genetic editing tools are available, and these are difficult to transpose to tailored individual studies. More specifically, for live cell imaging of organoids, fluorescence-based strategies are possible[39,65–67] but cumbersome, expensive to implement for patient-derived organoids, and restricted to a live imaging period of a couple of days at most[52]. The integration of fluorophores is also incompatible with cell therapy and organoid grafting therapeutic options. As a result, non-invasive, label-free volumetric optical microscopy appears to be a more natural solution for live cell imaging of organoids and other similar 3D tissues[23,49,68].

Label-free microscopy consists of using the intrinsic optical properties of biological structures, such as scattering, absorption, and phase contrasts, instead of external probes like fluorophores and monitoring how they change dynamically in order to characterize samples[69–71]. Recently, the analysis of the temporal dynamics of such endogenous contrasts enabled an important step to be made toward increasing the specificity of label-free microscopies[23,49,72].

Among other types of label-free microscopies[73], full-field optical coherence tomography (FFOCT) appears particularly suited to live cell imaging of organoids[49] thanks to its high 3D spatial resolution, lack of phototoxicity, contrast based on back-scattering and phase differences[8,12,13], high sensitivity, and high imaging speed[5,8,12–14]. Furthermore, temporal quantification of FF-OCT signal over a short time scale, named dynamic FF-OCT (D-FFOCT), has emerged as an invaluable metric of metabolic contrast[8,49,68,72] which has since been linked to subcellular organelle activity[14]. As a result, dynamic and static FF-OCT ((D)-FFOCT) have been used for in vitro and ex vivo studies to specifically resolve most cell types in 3D tissues/organisms[8,9,14,74,75], to identify different cell stage states such as senescence and mitosis[10,26,28], and to detect subcellular compartments and organelles[28]. D-FFOCT is, therefore, an appealing microscopy design to drive biology research on unaltered samples at high resolution and perform live cell imaging in thick samples, including 3D cell cultures.

However, whilst D-FFOCT has been successfully applied in retinal cell cultures such as young non-laminated retinal organoids[49], retinal explants[68], and retinal pigment epithelium[14,68], the lack of precise environmental control (temperature, $CO_2$ level, culture medium composition) on the research setups employed have up to now limited live cell imaging to under 3 h on the same sample[21,37–39,45].

In this work, for the first time, we designed a D-FFOCT module coupled to one of the optical ports of a commercial inverted microscope (IX83, Olympus, Japan). This design aims to standardize D-FFOCT make it more accessible to a large community, and facilitate its coupling with a wide variety of other imaging methods available on commercial microscopes. The design presented here could easily be adapted to other microscope brands and models with an accessible optical port by choosing appropriate alternate interfacing optics. Our microscope was directly installed inside a level 2 biosafety laboratory (L2) so that patient-derived cell cultures could be imaged over long periods of time. More importantly, we took advantage of this integration in a commercial microscope to demonstrate the use of D-FFOCT in physiological and environmental conditions thanks to a stage-top incubator. Although DFFOCT can be used in other types of samples, we chose here to focus on retinal organoids in order to

compare the technical performance of the new module with the state-of-the-art[49]. The use of a motorized translation stage as well as the automation capability of a commercial microscope, enabled us to perform an automatic continuous live cell imaging of an entire retinal organoid over more than 2 weeks without observing any sign of stress[14]. The retinal organoid was still alive at the end of this period of time, and processes such as rosette formation and mitotic proliferation were therefore followed live. Finally, we demonstrate a new method for imaging retinal organoids larger than 1 mm², older than 250 days, at a stage when photoreceptor cells are fully differentiated with outer segments present. Our ability to image these large organoids demonstrates the increase of our axial and transverse imaging ranges by factors of 2 and 4, respectively, compared to prior state-of-the-art[49].

## Results

**Volumetric D-FFOCT imaging of retinal organoids**. A volumetric acquisition was carried out on a 28-day-old retinal organoid (d28), with the newly developed D-FFOCT module (see "Methods"), inside a stage top incubator, as displayed in Fig. 1, with sufficient coverage to observe the entire organoid (Fig. 1a) whilst keeping a subcellular resolution, both in lateral (Fig. 1b) and axial directions (Fig. 1b–d).

Three dynamic metrics were calculated from a time series of 512 FFOCT images (see "Methods") and were displayed in a hue-saturation-brightness space (HSB)[49]. Red hue indicates faster activity than blue; saturation indicates activity randomness; and brightness captures the axial amplitude or strength of the activity. Using a different acquisition architecture than in prior work[49], the acquisition of 512 images at 100 Hz (5.12 s), data transfer (0.79 s), post processing (1.34 s), and saving (0.53 s) are parallelized, resulting in an effective total time of acquisition of 5.12 s per D-FFOCT image (see "Methods"), resulting in a time gain factor of 10 compared to the prior state of the art (see "Methods")[49]. Thanks to this acquisition speed, three-dimensional (3D) volumetric imaging is carried out on the d28 organoid, covering $400 \times 400 \times 120 \ \mu m^3$ in 2 h and 10 min, with a voxel size of $139 \times 139 \times 1000 \ nm^3$ (see "Methods"). 3D volume rendering is performed, as displayed in Fig.1f, g. We note that the total time of acquisition also included 2-phase static FFOCT acquisition and stage translation steps, accounting for a slightly longer acquisition time than 5.12 s per field of view. Each raw data stack is 3GB, while processed data can be reduced to <1GB per stack for storage.

Retinal organoids derived from human iPSCs are self-forming structures, initially composed of retinal progenitor cells (RPCs) that recapitulate human retinogenesis through the formation of stratified and organized retinal structures that display markers of typical retinal cell types[76–78]. Label-free volumetric images confirmed the self-organization of the structures into a neuroepithelium containing the cell bodies of aligned RPCs (Fig.1b, c), as previously described in studies using classical immunochemistry[76–78]. At d28, the RPCs appear as confluent green/yellow cells presenting a dynamic profile between 5.5 and 8 Hz in our culture condition, with a radial symmetry meaning that different cell shapes can be observed (see Supplementary Note and Fig.1).

**Longitudinal and volumetric D-FFOCT imaging of retinal organoids**. A longitudinal volumetric acquisition, with the newly developed D-FFOCT module (see "Methods"), was carried out on a single retinal organoid placed inside a stagetop incubator over 17 days up to d42. No sign of disturbance or abnormality was observed during the entire acquisition that would indicate cellular stress in the organoid[14,49], see Fig. 2.

A D-FFOCT volume, as previously demonstrated and highlighted in Fig. 1, was acquired each day with sufficient coverage to observe the entire organoid (Fig. 2) whilst keeping a subcellular resolution, both in lateral and axial directions (voxel size of $139 \times 139 \times 1000 \ nm^3$). The volumetric acquisition ranged from 2 h and 10 min for d28 to 8 h 32 min for d42 due to the size increase.

In order to perform the proof of principle regarding the use of D-FFOCT for time-lapse across several days and weeks, the organoid was embedded in 0.3% Matrigel (see "Methods") to keep the same orientation and "face" imaged, and hence view a similar cell organization (see Supplementary Note and Fig. 1) throughout the time-lapse. As a result, the position of the organoid was perfectly retrieved after removal and re-insertion of the multiwell plate on the microscope during medium changes. As a consequence, the stage-top incubator on the microscope does not necessarily need to host the sample during the whole period of the time-lapse, which can potentially be stored in a separate incubator, freeing up the microscope for carrying out other experiments. The organoid remained in a standard multiwell plate with its unsealed lid during the acquisition, which was placed in a stage top incubator (see "Methods") set at 37 °C, 5% $CO_2$, 20.5% $O_2$, 74.5% $N_2$, and 95% humidity. Medium change was carried out every two days under a fume hood placed in a level 2 (L2) biosafety laboratory environment, where the setup itself was installed to ensure optimal culture conditions. As a result, contaminations were kept to a minimum, to the standard of L2 and organoid production protocols, throughout the whole experiment.

A selected plane at 50 µm depth originating from the daily volume acquisition on the same organoid over 17 days is displayed in Fig. 2. A second plane at 4 µm depth is displayed in Supplementary Fig. 2 to allow comparison, as described in Supplementary Note 2. The evolution of cells and structures could be followed up over several days at subcellular resolution in the same organoid. Growth of the organoid occurs on a daily basis, thanks to the relatively low concentration of Matrigel embedding the organoid. The shape of the organoid evolves from a spherical ball of retinal progenitors, at d28, to a non-descript shape with both long and spherical rosettes, forming from d36, which reflects the proliferation of retinal progenitors leading to an increase of neuroepithelial tissue size observed in retinal organoids[76].

During the first week (d28–d35), the RPCs that compose the retinal organoid at d28 grow to form the neuroepithelial tissue at the periphery of the retinal organoid (Fig. 2). Within the two first weeks, rosette formations take place in the neuroepithelium as described previously[76]. We imaged cell motility that leads to the formation of the rosette structures in retinal organoids (Fig. 3a). As a result, we could track in 3D their formation and migration over time. We showed that the rosette lumen came from the reorganization of the peripheral layer of the neuroepithelium, implicating hinge regions, as observed in retinal organoids passing from the optic vesicle to the optic cup stage[79]. As a result, proliferative RPCs generated rosette structures which were included in the neuroepithelium over time (Fig. 3a).

The mitotic status of RPCs was highlighted by larger spherical cells than regular RPCs, displaying a red and green hue (5.5–13 Hz) close to the lumen, as shown in the d33 retinal organoids (Fig. 3a) surrounding the rosette, corresponding to a specific phase of the mitosis. In fact, D-FFOCT resolved cells in the replication/division stage, as shown in Fig. 3b–d, with a single cell passing successively from the prometaphase to the anaphase to the telophase over a time period of 16 minutes. Also, starting from d33 around the rosette, low-frequency blue filaments at 3 Hz can be observed. Similar larger filaments can also be

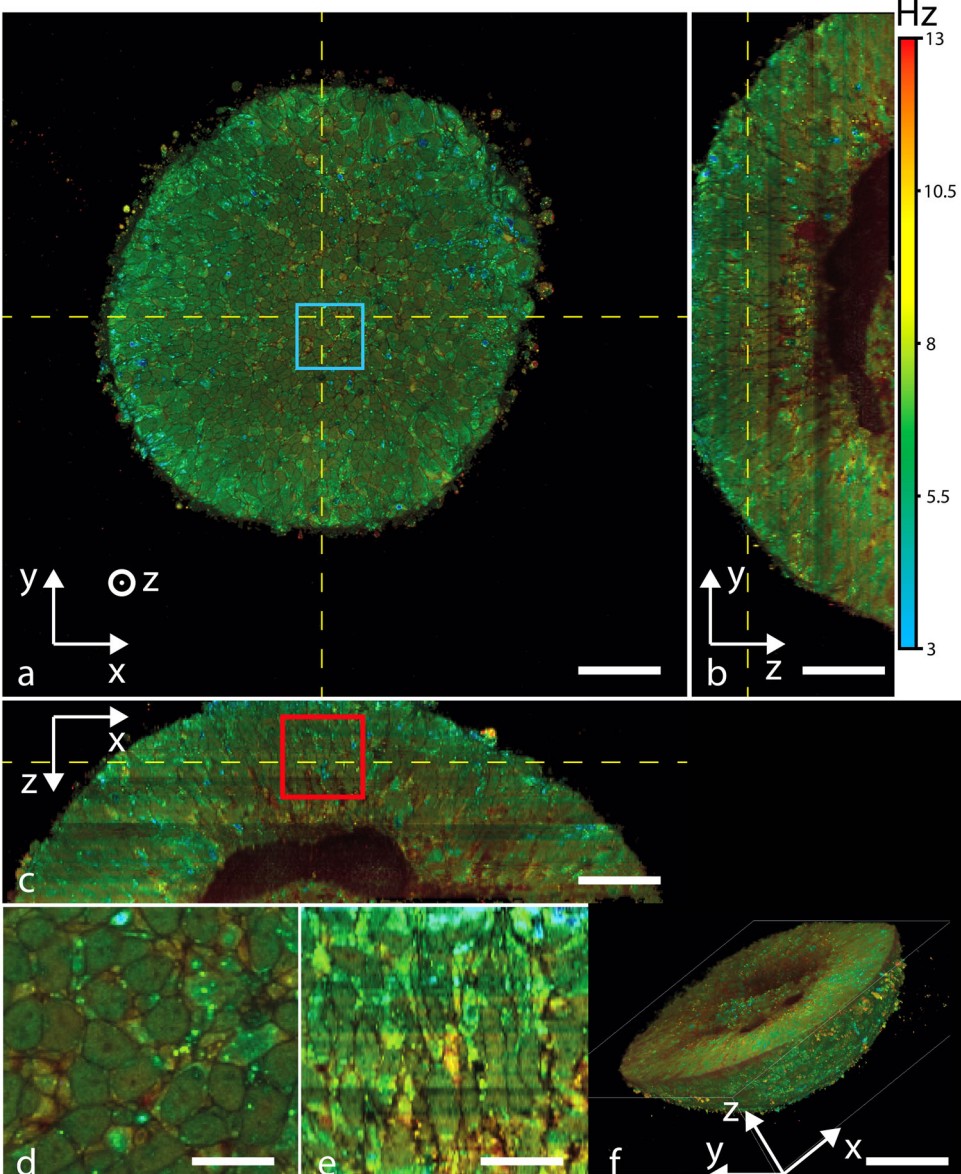

**Fig. 1 Example of D-FFOCT data acquired for each time point of the time-lapse.** A D-FFOCT 3 × 3 mosaic z-stack of a retinal organoid at d28 with an axial step of 1 μm is acquired in 150 min. Overlap of the tiles is 50%. Hue scales from 3 to 13 Hz mean frequency. **a** An *en face* (XY) cross-section from within the organoid, at 35 μm depth. A zoom-in on a, indicated by the blue square, is displayed in (**b**), where each cell can be resolved. **c** A vertical (YZ) cross-section from within the organoid. The vertical yellow dashed lines, displayed in (**a**, **c**), coincide. **d** A vertical (XZ) cross-section from within the organoid. The horizontal yellow dashed lines, displayed in (**a**, **d**), coincide. A zoom-in on **d**, indicated by the red square, is displayed in (**e**), where each cell can be resolved along the axial direction. **f**, **g** Reconstructed volumetric views of the same organoid. Scale bar, 50 μm in (**a–c**); 10 μm in d; 12.5 μm in (**e**) and 100 μm in (**f**).

observed from d35 onwards in Fig. 2. These filaments should theoretically correspond to retinal ganglion cell (RGC) axons, which is coherent with the fact that RGC is known to emerge in retinal organoids from d29[76]. We could also observe that such filaments appeared earlier at the bottom of the organoid, for example, at 4 μm above the glass coverslip, where we could detect them from d29 (see Supplementary Notes and Figs. 2 and 3). This earlier detection of the axon-like structures at the bottom of the retinal organoid, but not on the sides of the equatorial plane, may indicate that their formation is favored by the glass bottom of the multiwall plate, on which the organoid was resting (see "Methods"). Low-depth prolongation of the axon structures was also favored because the retinal organoids were placed in Matrigel[47].

Interestingly, whole images of retinal organoids between d35 and d42 (Fig. 2), generated by the D-FFOCT module, showed large rosette structures with long lumen (Fig. 4c), where RPCs are still proliferative, surrounded by the emerging RGCs and retinal inner neurons as described in retinal organoids[76] and shown in Fig. 5b. Similar observations were made on different retinal organoids from the same batch on which the longitudinal experiments were carried out. Small "punctual" rosettes started to appear from d32, mostly from the periphery of the retinal organoid, as highlighted in Fig. 3, and long rosettes started to appear from d33, as displayed in Fig. 2 and highlighted in Fig. 4. The main axis of the cells surrounding the long rosettes was normal to the lumen axis, as shown by the similarity between Fig. 1e and Fig. 4b.

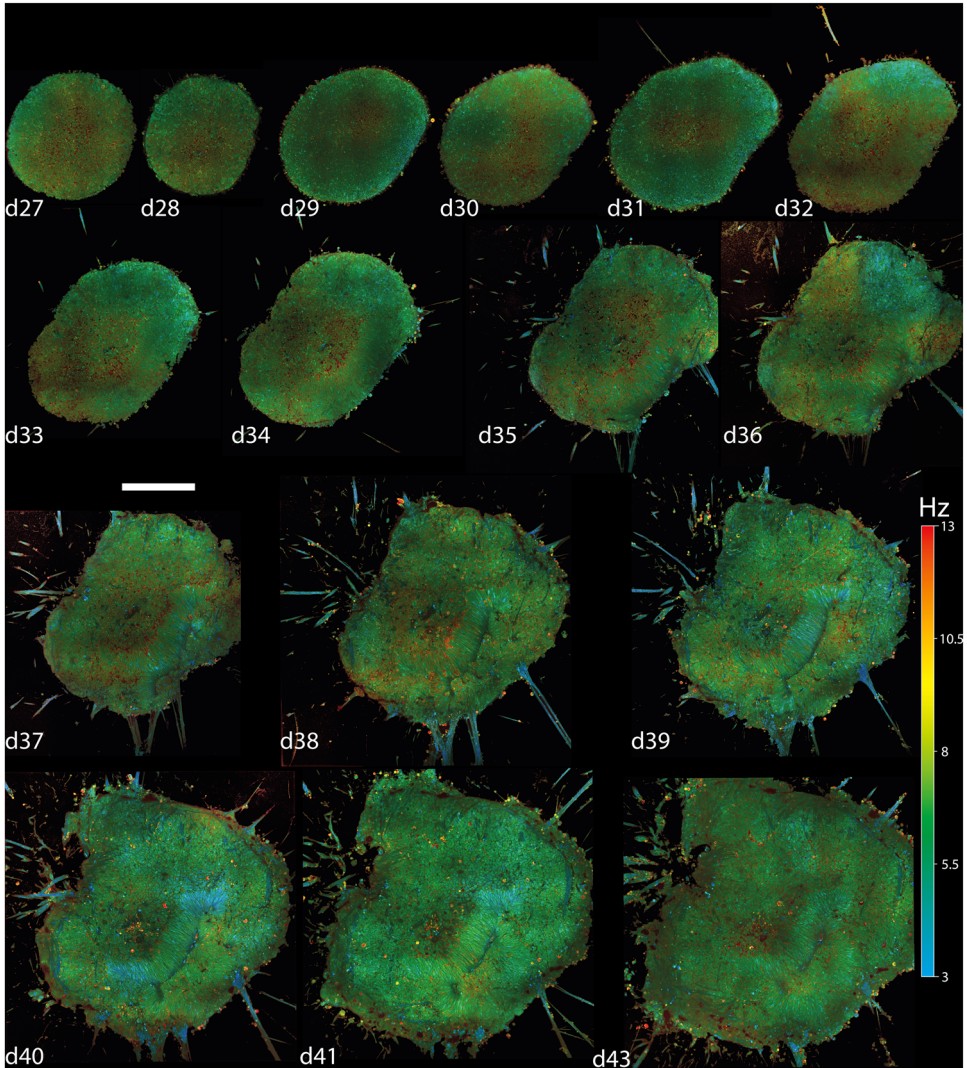

**Fig. 2 D-FFOCT volumetric and longitudinal imaging of one single organoid at a depth of 50 μm across 17 days.** Hue scales from 3 to 13 Hz mean frequency. The evolution of cells and structures could be followed up each day at subcellular resolution in the same organoid. The organoid grows daily, evolving from a spherical ball of retinal progenitors at d28 to a non-descript shape with both long and spherical rosettes, forming from d36, which reflects the proliferation of retinal progenitors leading to an increase of neuroepithelial tissue. Image mosaicking covers 406 × 406 μm², 2928 × 2928 pixels, (3 × 3), at day 28 (d28) to 717 × 717 μm², 5163 × 5163 pixels, (6 × 6), at day 42 (d42). The scale bar represents 50 μm and stands for all panels. Scale bar, 50 μm.

Axonal extensions started to be generated inwardly in retinal organoids from d33 (Fig. 2). Belonging to RGCs, which are the first differentiated neural cells in retinal organoids[80], axonal elongations were resolved with D-FFOCT at high resolution, showing long associated nerve fibers that can compose the optic nerve in vivo (Fig. 4d, e).

The temporal evolution of long rosettes could also be followed over time, with an epithelium organization highlighted with dashed lines in Fig. 5. Interestingly, the main cell axis of the RPCs was observed to be radial to the rosette. If RPCs can be described analogously to a 3D ellipse, it means that their major axis is included in the imaging plane. By symmetry, imaging along this axis recapitulates the different aspects of an RPC in a single imaging plane. By extension, since the neuroepithelium is also growing radially to the rosettes, imaging this tissue along their main axis enables recapitulation of both RPC and neuroepithelium fate. Therefore, the alignment of the main cell axis with the D-FFOCT imaging plane enables visualization in a single image of the tissue mesoscopic architecture whilst resolving its cell units.

This is a more effective way to monitor retinal organoid epithelium as it requires only a single plane rather than a volume to decipher the retinal organoid epithelium state.

In order to demonstrate faster-paced longitudinal acquisition, a time-lapse on a locked plane was carried out over 11 h, with mosaic imaging (3 × 3) and a larger time series of 1024 FFOCT images (see "Methods"), producing a reconstructed D-FFOCT image every 100 s (Supplementary Movie, showing timelapse over 11 h on a locked plane at 50 μm depth, field size 406 μm × 406 μm, on the organoid from Fig. 4). This ability shows that faster time monitoring could be conducted on a single plane. However, D-FFOCT single-plane imaging at 50 μm depth does not encompass the general state of the organoid, as mentioned earlier. Therefore, imaging at greater depth within more conventional retinal organoids, with proper layering, would statistically enable the capture of more cells in a radial position. This principle is highlighted in Fig. 4b and Fig. 5, where radial cells normal to rosettes can be observed, for example.

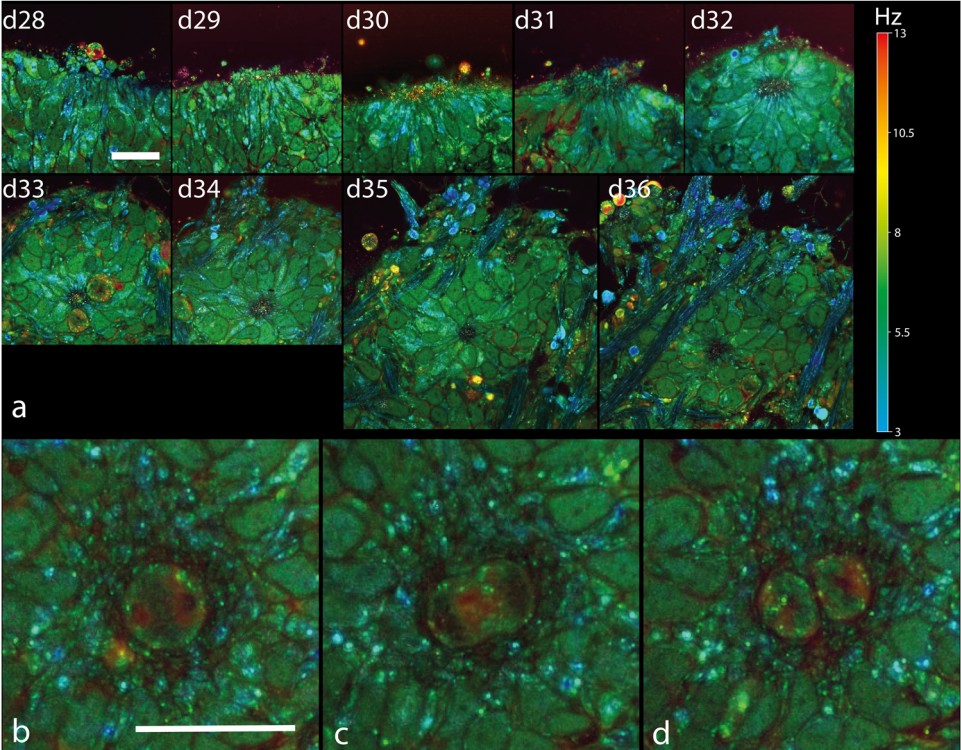

**Fig. 3 Tracking rosette formation and mitosis.** D-FFOCT longitudinal time tracking of a rosette formation over 8 days (**a**) and mitosis of a retinal progenitor cell over 16 min (**b**–**d**) Hue scales from 3 to 13 Hz mean frequency. **a** A single rosette found in the large volume displayed in Fig. 2 was tracked throughout each daily volume imaged. All images share the same scaling, with images from day 28 to 34 covering 70 × 70 μm², 500 × 500 pixels, and images from day 35 to 36 covering 105 × 105 μm², 750 × 750 pixels. **b** Highlights a spherical cell in a prometaphase with two spindle poles. **c** Highlights the same cell, 8 min later, which has become elongated, characteristic of the anaphase. **d** Highlights the two resulting daughter cells, 8 min later again, after the telophase. Images cover 40 × 40 μm², 290 × 290 pixels. An 8 min separates each image. Scale bar, 20 μm.

To confirm the integrity of retinal organoids during long-itudinal acquisition, we performed detection and quantification of apoptosis at single cell level, based on labeling of DNA strand breaks (TUNEL experiments, Fig. 6) and immunostaining triggering retinal cell population. In situ cell death analysis coupled with immunostaining confirmed that illuminated d28, d35, and d42 retinal organoids did not present more apoptotic cells compared to those that were not illuminated (Fig. 6a, c, e). At d28, d35, and d42, TUNEL+ cells within non-illuminated retinal organoids represent 2.69% +/− 0.62, 1.74% +/− 0.83, and 2.43% +/− 0.64 and within illuminated retinal organoids 4.17% +/− 0.19, 1.81% +/− 1.2, and 2.63 +/− 2.03, respectively (Fig. 6b, d, f, Supplementary Data). The number of retinal progenitors (VSX2+/PAX6+ cells), as well as the first differ-entiated retinal cells (PAX6+/VSX2−), were not impacted. In addition, we confirmed that during image acquisition, the light used for D-FFOCT did not impact the differentiation commit-ment of progenitors to the RGC population (BRN3A/PAX6+ cells, Fig. 7).

**Improvement of D-FFOCT imaging range for retinal orga-noids.** D-FFOCT imaging of retinal organoids at high resolution, using objectives with a numerical aperture (NA) higher than 0.8, has been limited to a depth of 80 μm, using a source at 660 nm, in previous work[49]. However, conventional free-floating retinal organoids are spheres that quickly reach diameters >700 μm during their growth[76]. Since retinal organoids replicate long-range (>100 μm) cell gradients within the retina[25,76], it is thus critical to be able to image up to a minimal depth of 100 μm in order to capture the overall cell-type organization in organoids, or

mesoscale features, using D-FFOCT[47]. Although we show D-FFOCT images from a retinal organoid at 120 μm depths, reaching the hollow center of the organoid using a light source at 730 nm (see Fig. 1c, d) requires a significant amount of time to acquire a volumetric dataset, enabling observation of mesoscale features, as displayed in Fig. 1e. Furthermore, D-FFOCT images at >100 μm depth arguably lack sufficient SNR to resolve each individual cell and their subcellular features. Ideally, in order to capture both mesoscale tissue cell organization and subcellular features in a minimum amount of time, imaging would need to occur at the "equator" of these spherical organoids, where the radial structures of the organoid align with the *en face* D-FFOCT imaging plane (see Supplementary Note 1). As a result, higher SNR is needed to optimize the data acquisition on retinal orga-noids using D-FFOCT, as well as mosaicking over 1 mm² areas on free-floating organoids.

In this section, we demonstrate *en face* imaging of a free-floating organoid over 1 mm², up to 230 μm depth, on retinal organoids older than 250 days, using a source with a longer wavelength of 810 nm and higher power (20 mW), resulting in an increase of SNR (see "Methods"). Optimal sampling frequency has been established in the past at 100 Hz for retinal organoids, with a lower D-FFOCT signal observed at higher or lower sampling frequency[49]. However, here, we used an acquisition rate of 500 Hz while maintaining the camera full-well-capacity (FWC) near maximum, which we recast in an effective time series at 100 Hz by performing a temporal binning in groups of 5 successive FFOCT images. As a result, the SNR of the time series was increased by a theoretical factor of $\sqrt{5}$. Therefore, longer wavelengths and temporal binning of the data lead to

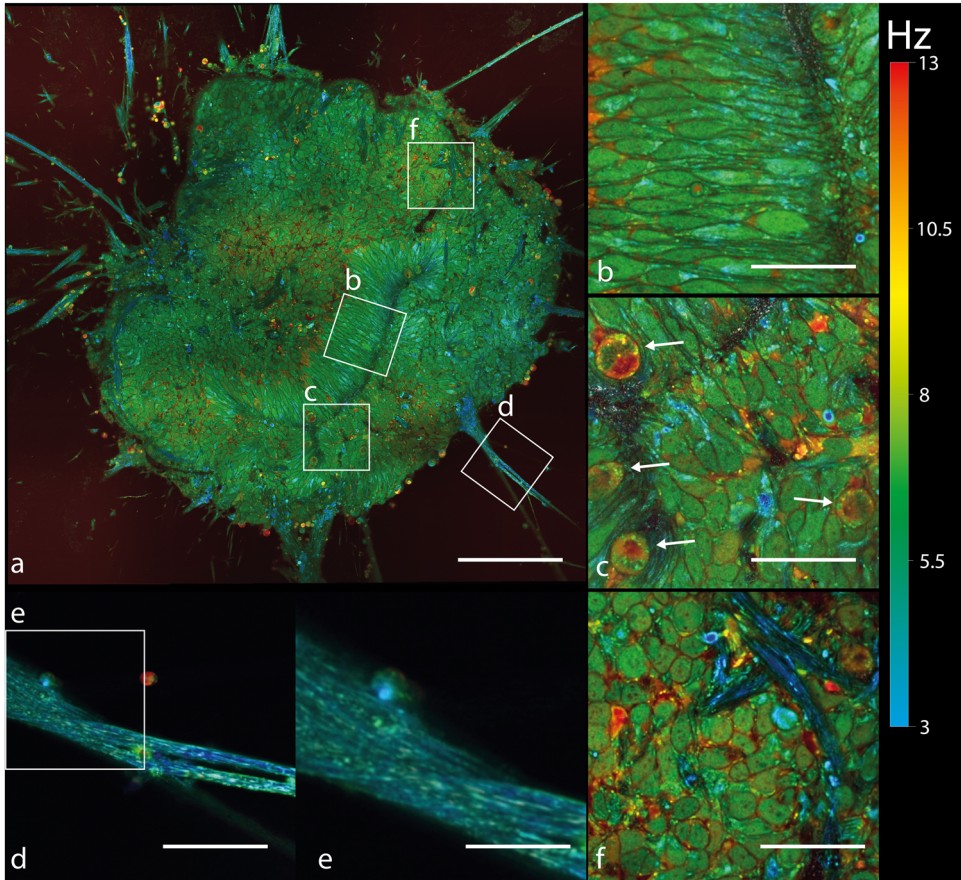

**Fig. 4 Highlights different structures of a retinal organoid (d38) using D-FFOCT.** Hue scales from 3 to 13 Hz mean frequency. **a** A single plane mosaic (6 × 6) of 717 × 717 µm², 5163 × 5163 pixels, scale bar 140 µm. **b–d, f** Magnified views of a of 79 × 79 µm², 570 × 570 pixels, scale bar 30 µm. **e** A magnified view from (**d**) covering 35×35 µm², 250 × 250 pixels, scale bar 11 µm. **b** Highlights normally-oriented retinal progenitor cells (RPCs), which are paraxial to the imaging plane. If the RPC can be described analogously to a 3D ellipse, it means that its major axis is included in the imaging plane. **c** Highlights mitotic RPCs located near the long rosettes. White arrows in **c** point out mitotic RPCs. **d** A retinal ganglion cell axon-like structure extending outwardly from the retinal organoid. **e** Highlights the fact that our setup can resolve single fibers composing this axon-like structure. **f** Highlights axon-like structures from retinal ganglion cells propagated inside the organoid.

higher SNR levels in organoids, enabling imaging at depths up to 230 µm.

Using this acquisition scheme, we were able to image mesoscale tissue features (Fig. 8a, b) and sub-cellular details (Fig. 8c–f) of much older retinal organoids on a single D-FFOCT mosaic (10 × 10) image in 12 min. In this type of acquisition, different dynamic profiles can be observed at the sub-cellular level (Fig. 8c–f). As previously reported by fixed organoid approaches (whole mount or cryosections[78,81]), our strategy highlights the layered organization of the organoid cells with different cell types (Fig. 8a, b). This approach allows the detection of long filaments of approximately 50 µm in length, located around the edge of the organoid (Fig. 8b, f), which likely correspond to the inner and outer segments of the photoreceptors. The layer of the retinal organoid adjacent to these filament structures towards the outside of the retinal organoid, which corresponds to the putative photoreceptor outer nuclear layer, is well-defined with very distinguishable cells (Fig. 8c). This layer seems delimited towards the internal part of the organoid by a putative outer plexiform layer, mostly present in the whole observation plane (Fig. 8a, b). On moving further to the interior of the organoid, this is followed by cells presenting different mesoscale features composed of yellow nucleus cells and larger red cells (Fig. 8d), which may correspond to the soma of bipolar cells and Müller glial cells that were previously identified by immunostaining in organoids of the

same age[78,81]. We note that the cells in Fig. 8c, d, contained in different retinal layers, are of similar morphology but are nevertheless more easily differentiable thanks to the behavioral information rendered in the H and S channels of the color map: the differing frequencies of subcellular motion within these cells show one type to be moving at a faster frequency with at a narrower bandwidth, which can act as another label-free indicator of their identity as previously shown[49]. Finally, in the innermost part of the organoid, distinctive speckled and saturated yellow cells appear (Fig. 9e) that correspond to dying cells, as previously described within the center of large organoids[81,82] and established with D-FFOCT[14].

Although not imaged strictly at the "equator" of the organoid, D-FFOCT mesoscale imaging was significantly improved compared to D-FFOCT imaging at lower depths. An example of imaging limited to 80 µm depth in a 160-day-old retinal organoid is shown in Supplementary Note and Fig. 4. As a result, retinal organoids can be monitored at a faster rate than with volumetric imaging while encompassing the general state of the retinal organoid (see Supplementary Note 1).

**Imaging in ex vivo tissues.** To evaluate the performance of D-FFOCT to image the highly characteristic physiological organization of photoreceptors, ex vivo adult pig retinal explants were

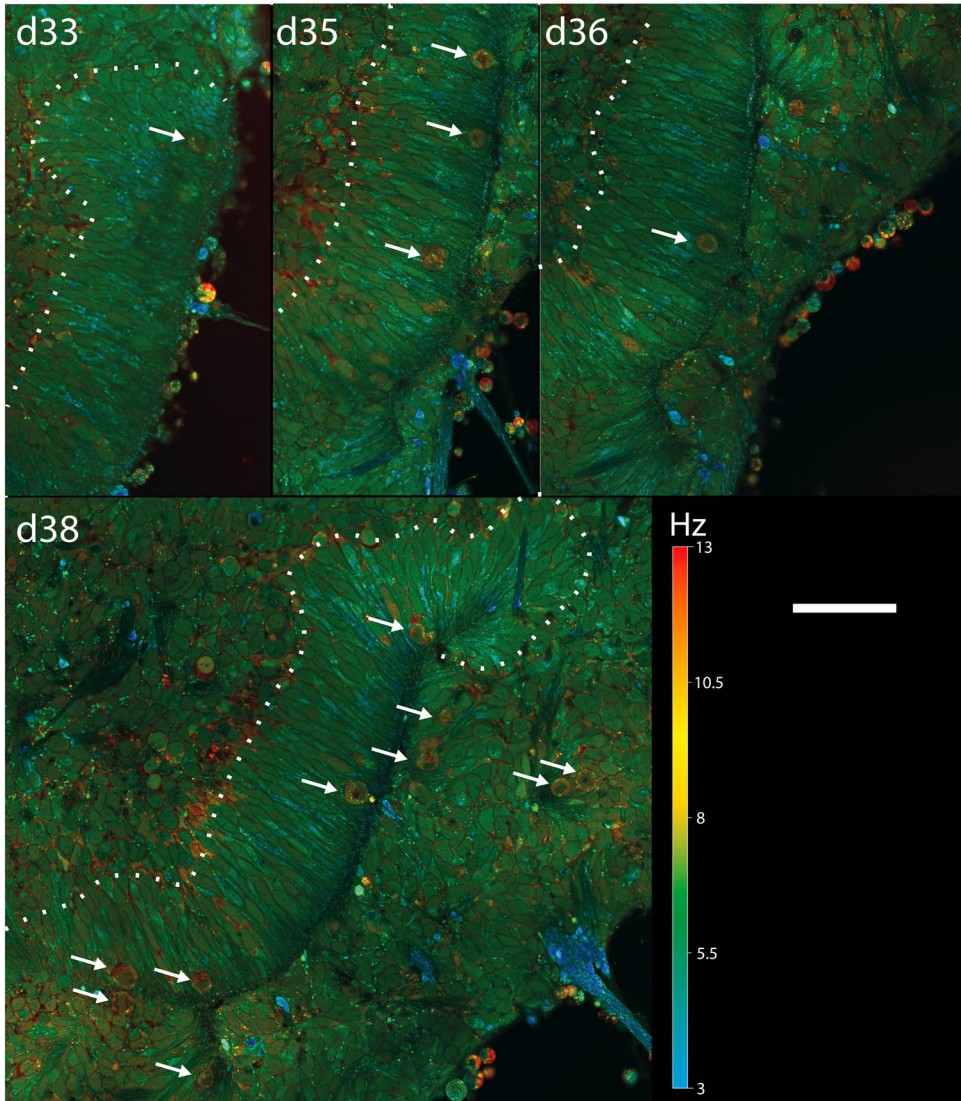

**Fig. 5 D-FFOCT longitudinal time tracking of a long rosette formation over a couple of days.** Hue scales from 3 to 13 Hz mean frequency. A single long rosette was tracked throughout each daily volume imaged. All images share the same scaling, with images from d33 to d34 covering 128 × 238 µm², 924 × 1713 pixels, and images from d36 covering 205 × 273 µm², 1480 × 1968 pixels, and d38 covering 313 × 296 µm², 2252 × 2128 pixels. White arrows highlight mitotic RPCs. Scale bar, 50 µm.

also imaged under culture conditions. Focusing on the photoreceptor layer, images obtained clearly show the cone and rod photoreceptor mosaic in *en face* slices, with subcellular detail of mitochondria visible inside the cell bodies and the reconstructed depth slice revealing photoreceptor shape including the characteristic cone inner and outer segments (Fig. 9).

## Discussion
Live imaging with D-FFOCT achieves non-invasive label-free 3D viewing of in vitro and ex vivo samples. In comparison to other label-free microscopies, D-FFOCT provides imaging in depth (in contrast to differential interference contrast microscopy), no measurable phototoxicity (Figs. 6 and 7) and fast acquisition time (in contrast to multiphoton microscopies such as third harmonic generation and coherent anti-Raman Stokes microscopies) and crucially, the dynamic aspect of D-FFOCT provides behavioral information which is inaccessible to these other methods, in addition to revealing the 3D structure. In this work, we have demonstrated a D-FFOCT module design coupled to a

commercial microscope with a stagetop incubator, which allowed 3D, longitudinal imaging in retinal organoids over periods of weeks. Organoids are highly promising for the fields of disease modeling, drug development, and gene and cell therapies[3–5]. To effectively image organoids, we sought to develop a technology that could meet the requirements of high-resolution, long-term, live, volumetric imaging, with improved performance and accessibility compared to our previous device[49]. The use of longer wavelength LED illumination and time series binning allowed acquisition at deeper penetration depths than had been previously demonstrated[49], and the use of the commercial microscope's translation stage allowed coverage of areas larger than 1 mm² while conserving a theoretical lateral spatial resolution <400 nm[10]. These two capabilities open up D-FFOCT applications on thicker and larger samples, as well as the imaging of retinal organoids in a more equatorial plane. This last aspect is especially interesting as it enables simultaneous monitoring of all the cell layers of the retinal organoid at high resolution, therefore providing an overview of the organoid's overall state in a single D-FFOCT mosaic plane rather than having to make a volumetric

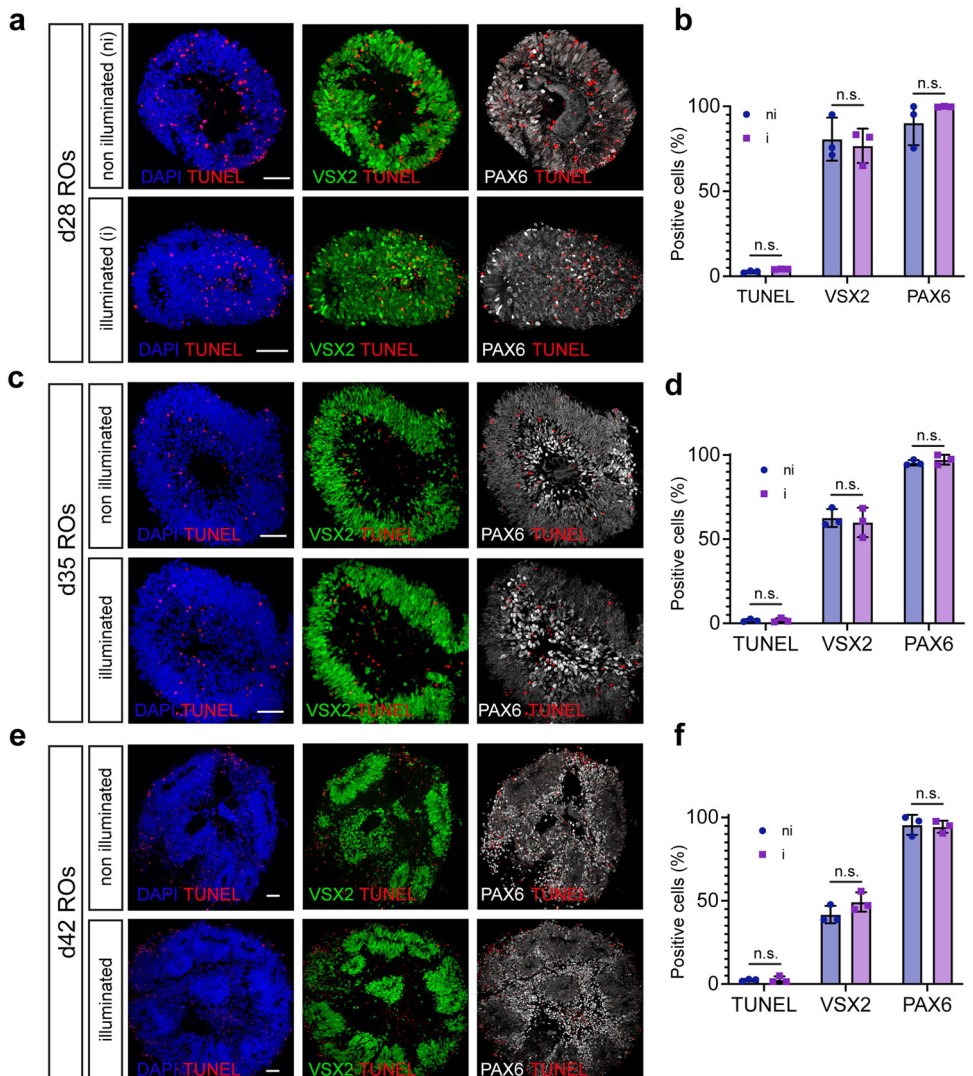

**Fig. 6 Cell population analysis within non-illuminated or illuminated d28, d35, and d42 retinal organoids. a** Staining of apoptotic cells (TUNEL assay, red), retinal progenitors (VSX2+, green) or total cell population (progenitors PAX6+/VSX2+ and the first differentiated retinal cells, PAX6+/VSX2−, white) in d28 retinal organoids. **b** Quantification of TUNEL, VSX2 and PAX6 positive cells in d28 retinal organoids. **c** Staining of apoptotic cells (TUNEL assay, red), retinal progenitors (VSX2+, green) or total cell population (progenitors PAX6+/VSX2+ and the first differentiated retinal cells, PAX6+/VSX2−, white) in d35 retinal organoids. **d** Quantification of TUNEL, VSX2, and PAX6 positive cells in d35 retinal organoids. **e** Staining of apoptotic cells (TUNEL assay, red), retinal progenitors (VSX2+, green) or total cell population (progenitors PAX6+/VSX2+ and the first differentiated retinal cells, PAX6+/VSX2−, white) in d42 retinal organoids. **f** Quantification of TUNEL, VSX2, and PAX6 positive cells in d42 retinal organoids. Data are presented as mean ± SD ($n = 3$). Mann–Whitney comparison test. Nuclei were counterstained with DAPI (blue). Scale bar, 50 µm.

acquisition, which implies longer acquisition times and data volume. As a result, control quality and general assessment can be achieved in a quicker way than volumetric acquisition. Live imaging can, therefore, also be conducted at a faster pace. Furthermore, we note that mosaics were reconstructed from tiles overlapping by a factor of 50%. This inefficient coverage is due to D-FFOCT signal inhomogeneities which occur with the high NA objectives used in D-FFOCT, as sample and reference coherence planes do not overlap perfectly in space. Additional degrees of freedom for controlling the reference mirror inclination (see "Methods") independently from the objective inclination would enable the correction of this defect. As a result, using an overlap of 10% instead of 50% for a mosaic becomes feasible. In this case, the area covered by a $10 \times 10$ mosaic can be accomplished by a $6 \times 6$ mosaic with 10% overlap leading to a threefold time gain. Furthermore, mature retinal organoids are empty in their center, meaning that this area could be skipped, reducing further the

number of tiles necessary to create a large-scale mosaic at this resolution. Additionally, active control of the reference time delay at high depth would benefit the mosaic image quality by having a more homogeneous contrast and higher global SNR. Indeed, the time delay between the reference arm and the sample arm (see "Methods") was not constant at a given depth for different $x,y$ coordinates. The heterogeneities in mature retinal organoids cumulated significantly at depths greater than 100 µm resulting in a significant optical path difference mismatch depending on the plane location, on the order of 1–2 µm. Finally, parallel management of data and optimization of post-processing algorithms enabled close to 10 times faster acquisitions than previously demonstrated[8,14,49,68], which is critical for imaging large samples, volumetric imaging, and achieving live imaging, while data storage is reduced to <1 GB per stack after processing.

The use of the commercial microscope's translation stage allowed accurate mosaicking over wide regions to cover larger

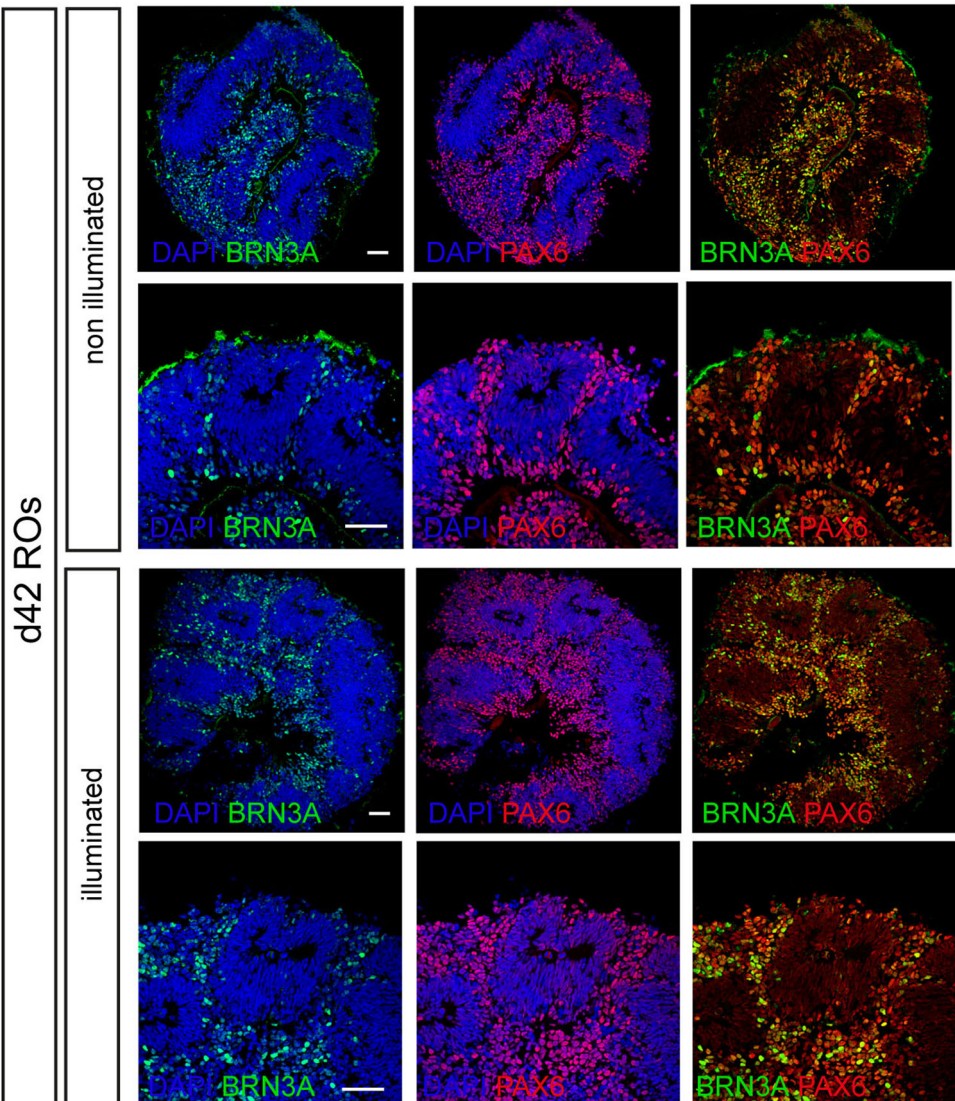

**Fig. 7 Retinal ganglion cell (RGC) population analysis within non-illuminated or illuminated d42 retinal organoids.** Immunostaining of RGCs (BRN3A +/PAX6) in retinal organoids at d42. Nuclei were counterstained with DAPI (blue). Scale bar, 50 μm.

organoids than previously demonstrated, and positioning was reproducible when moving from well to well to the extent that the same field could be recovered without the need for image registration when moving away from and back to the same sample or removing and resetting the plate in position for a subsequent acquisition when using embedding Matrigel. In the case where the sample is free-floating, slow stepping of 2 μm every 40 ms was carried out in order to minimize movement. No drift was monitored on retinal organoids older than 250 days, such as the one shown in Fig. 8. As a result, D-FFOCT can be used for monitoring organoid production using different protocols, both with or without embedding.

An additional demonstration of this D-FFOCT module was carried out on retinal explants with a focus on photoreceptors (Fig. 9) in order to show imaging performance in ex vivo tissues. To the best of our knowledge, these images represent the highest resolution on photoreceptors achieved by an OCT technique.

Where past work used a standalone D-FFOCT device under optics laboratory conditions to highlight 3D imaging in organoids[49] and cell behavior under stress in the context of disease modeling[14], here, our technical results extend the potential of this promising technique by adapting it to use by a wider

community of biologists and other non-optics experts. This was achieved by integrating D-FFOCT into a conventional microscope setup, familiar to biologists, thereby facilitating adoption; allowing continuous protection of the samples under controlled environmental conditions in a stage-top incubator suitable for culture; and by precisely automating key aspects of acquisitions, including repeatable $xyz$ positioning, 3D stack capture, and mosaicking; along with developing an efficient and fast workflow for acquiring, post-processing and saving datasets. In addition to these protocol improvements, in terms of optical innovation, we showed greater penetration depths achievable with longer wavelength illumination and the improved SNR gained by binning, which opens up the perspective of making faster acquisitions with fewer frames required to calculate dynamic metrics than in past work. Biological results shown include clear 3D views of cone and rod photoreceptors in retinal explants with an unprecedented resolution for an OCT-based technique (Fig. 9); two types of iPSC-derived retinal organoids with cell layering distinguishable from the dynamics and morphology of the D-FFOCT signal; and long term acquisitions on single organoids over >40 day periods, highlighting the live follow-up of normally inaccessible dynamic processes such as rosette formation (Figs. 3 and 5) and mitotic

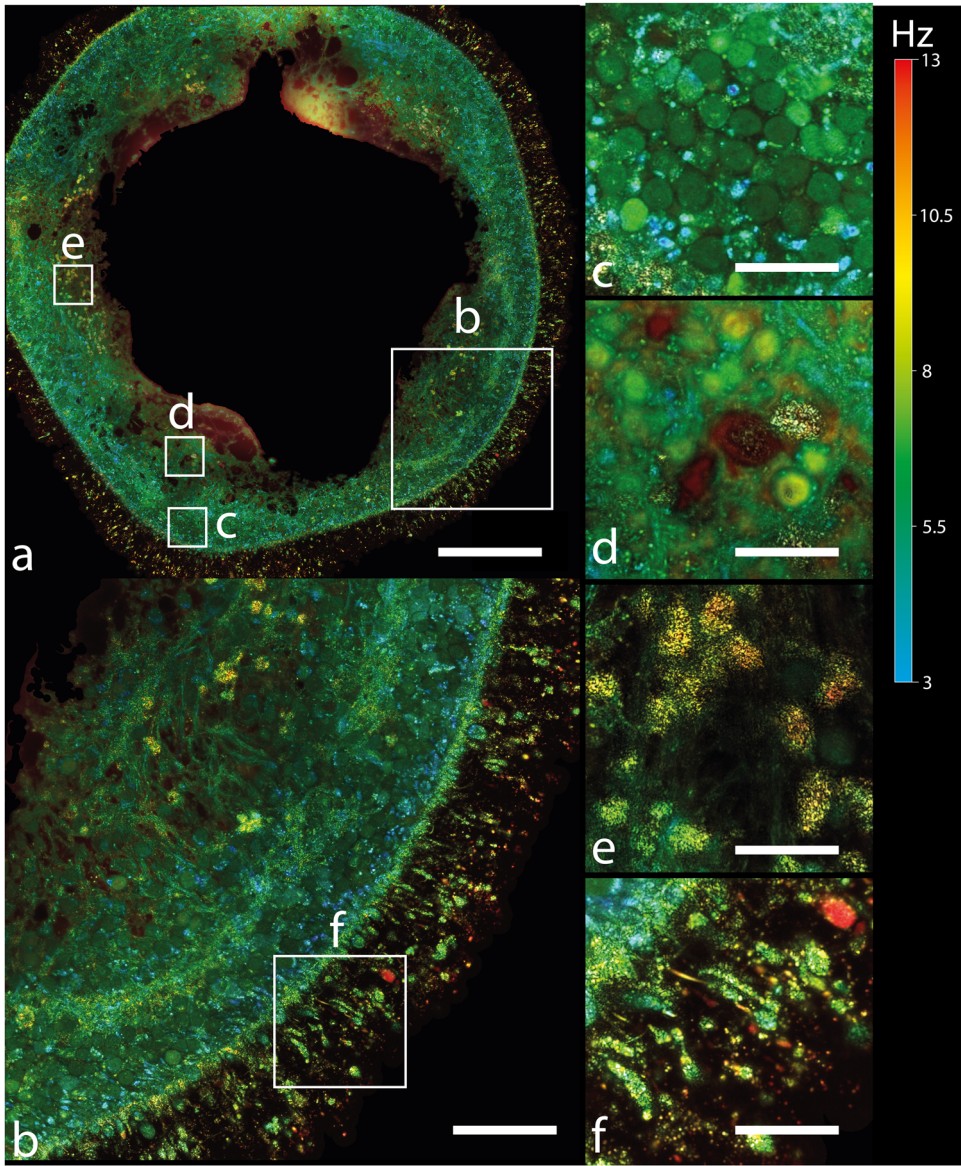

**Fig. 8 D-FFOCT large-scale mosaic imaging of a retinal organoid highlighting mesoscale and subcellular features. a** d266 retinal organoid imaged at 190 μm depth by D-FFOCT. 10 × 10 tile mosaic with 50% overlap, corresponding to 1.125 × 1.125 mm², 8101 × 8101 pixels. Hue scales from 3 to 13 Hz mean frequency. **b** Displays a zoom-in of **a**, highlighting the cell layering organization of the organoid with different cell types. Outer and inner segments of photoreceptors are located around the edge of the organoid (50 μm thick layer) (**a**, **b**, highlighted in (**f**). The outer nuclear layer of the photoreceptors is well-defined with very distinguishable cells, as highlighted in (**c**), and delimited by the outer plexiform layer, mostly present in the whole plane. Fiber-like structures follow up with yellow nucleus cells; **d** intertwined within these fibers; which may be bipolar cells. Finally, mostly in the inner part of the organoid, distinctive speckled and saturated cells appear, which may be dying or dead cells (**e**). Scale bar, 200 μm in (**a**), 60 μm in (**b**), 10 μm in (**c**), 25 μm in (**d–f**).

proliferation (Figs. 3 and 5), with the ability to probe mature organoids at later stages of development enabled by the wide field mosaicking and deeper penetration depths. We validated that D-FFOCT imaging does not interfere with the development of the organoids or induce cell death during longitudinal acquisition (Figs. 6 and 7). In follow-up work, we anticipate the use of the D-FFOCT module in disease modeling and drug screening applications in 3D samples, such as organoids and explants, as well as 2D samples, such as cultured cell sheets. A challenge currently being tackled is the efficient management of the large datasets generated by this high-resolution volumetric imaging method. Various post-processing methods are being explored to extract pertinent data and metrics in an efficient manner, including the use of machine learning-based methods to quantify cell density, morphometry, and dynamic profile in order to

facilitate automation of image analysis[9]. With increased automation of acquisition and image analysis, D-FFOCT may find its place as a tool of choice for live imaging in therapeutic screening and quality control trials of cultured tissues.

## Methods

**Optical setup.** A schematic of the optical setup is displayed in Fig. 10, with its components described. The D-FFOCT module is a Linnik interferometer with one additional lens in the sample arm and balanced in the reference arm (L3 and L4 in Fig. 10). This configuration enables an efficient and homogeneous illumination with a multiport microscope as the distance between the objective in the sample arm (Obj.1) and the microscope output port, where no optics can be placed without altering the microscope hub for other imaging techniques, is relatively important

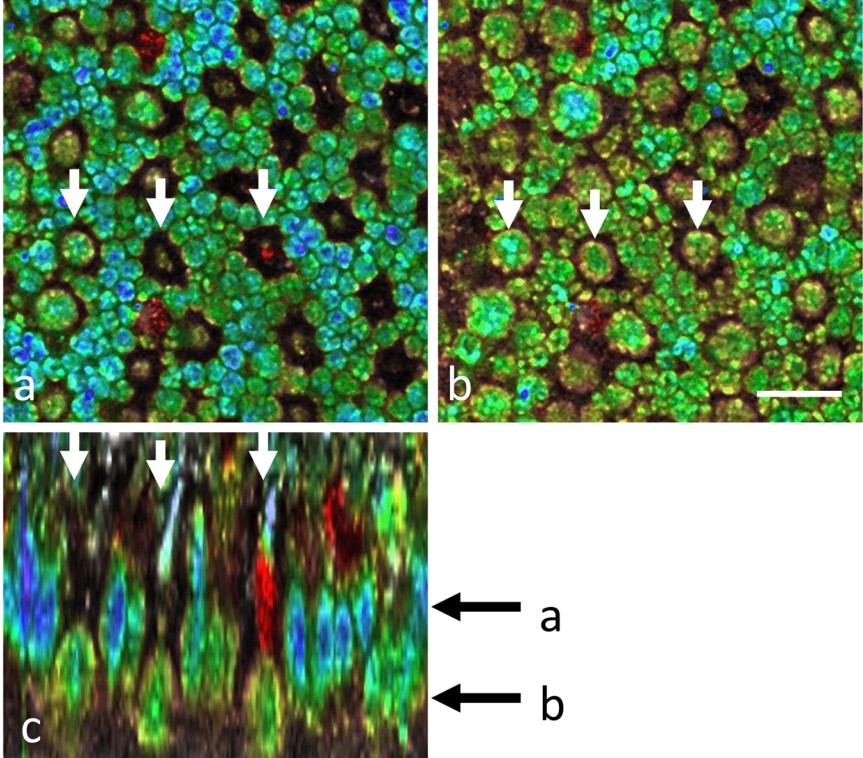

**Fig. 9 D-FFOCT in the photoreceptor layer of a porcine retinal explant, imaged under culture conditions.** The cone and rod photoreceptor mosaic were revealed in *en face* (**a**, **b**) and axial (**c**) slices. The depth positions of **a** and **b** are indicated by the black arrows next to (**c**). White arrows highlight three cones which can be visualized in all three views. Scale bar, 10 μm.

for a D-FFOCT optical layout[8]. This rather unconventional configuration allows the imaging of incoherent reflections, predominantly (99.9%) coming from the reflections on the non-polarizer-beam-splitter (NPBS) cube, in their Fourier plane located at the cMOS camera (Q-2HFW, Adimec, Netherland), thus reducing their contribution compared to a classical configuration[74]. The cMOS camera is a Q-2HFW (Adimec, Netherland), with a pixel size of 10 x 10 μm$^2$ and 1440 x 1440 pixels, and is set to obtain an SNR level of 1071. Two different LEDs were used in this work, with the central emission wavelengths at 730 μm (LED730) and 810 μm (LED810) (M730L5 and M810L3 respectively, Thorlabs, Newport, NJ, USA). The optical elements in the sample, reference, and detection arms give a magnification $M = 58$ and an imaging field of $200 \times 200$ μm$^2$, thus a pixel size of $139 \times 139$ nm$^2$. The power applied on the sample is 3.3 mW and 20 mW, resulting in values of intensity of 53 mW mm$^{-2}$ and 318 mW mm$^{-2}$, for LED730 and LED810, respectively. Note that although the power applied is similar to levels used in single photon fluorescence microscopy, the absence of excited fluorophores that create oxidative stress, the continuous wave nature of the source, and longer near-infrared wavelengths used here significantly reduce the danger of phototoxicity[66,83]. The exposure time is set at 3.9 ms or 1.3 ms to achieve 95% of the camera FWC for LED730 and LED810, respectively. In the case of LED730, a time series of 512 images is acquired at 100 Hz and is used to generate 3 metrics: the average of the power spectral density (PSD) frequency, the standard deviation of the PSD frequency, and the mean of the running standard deviation, with a sliding window of 50 elements, according to a methodology developed by Scholler et al. (2020), for contrast standardization as well as comparison[49]. In the case of LED810, a time series of 2560 images is acquired at 500 Hz and binned in groups of 5 consecutive frames before generating the same metrics as for LED730. The three dynamic metrics (mean frequency of the

PSD, standard deviation of PSD, and averaged running standard deviation) are combined in an HSB space, respectively.

**Optimization of data workflow (acquisition, postprocessing, and saving).** Acquisitions are triggered through hardware using a transistor-transistor logic of 0–5 V, generated by an acquisition card (NI cDAQ-9174, National Instruments, TX, USA), and data from the camera (11 bits) are transferred via 4 CoaxPress cables (25 Gbit/s) to a frame grabber (Cyton CXP4, Bitflow, Massachusetts, USA) which is connected to a PCIe 3.0 (32 Gbit/s) of a PC motherboard (WS X299 SAGE, ASUS, Taiwan). Data from the camera are converted to 16 bits by the frame grabber and are then transferred on a parallel thread of the random-access memory (RAM) (8x Vengeance LPX 1×16 GB 3000 MHZ, Corsair, California, USA) of the motherboard and logged onto this same thread. Using a multi-threaded architecture, the effective acquisition time $T_{tot}$ is bottlenecked to the irreducible time needed for data to be logged ($t_{logging}$). For example, for generating 10 D-FFOCT images (H,S,B), from $N_{batches} = 10$ batches, on a locked location, the total time of acquisition is $T^{multithread}_{tot} = 10 \times [N \times T_s] + t_p + t_{gpu} + t_{save}$ rather than $T_{tot} = 10 \times [N \times T_s + t_p + t_{gpu} + t_{save}]$ as in the previous state of the art[68], where $T_s$ is the inverse of the frame rate of acquisition, $N$ the number of images per batch, $t_p$ the time to transfer the data onto the computing RAM, $t_{gpu}$ the post-processing time on graphical processing unit (GPU), and $t_{save}$ the time it takes to save the post-processed data. The general workflow is sketched in Supplementary Fig. 5 (see Supplementary Note 5). As a result, continuous and lossless acquisitions are possible, and the effective generation of dynamic metrics tends ($N_{batches} \rightarrow \infty$) to the incompressible time it takes to acquire the frames themselves—as long as $[N \times T_s] \geq t_p$, and/or $[N \times T_s] + t_p \geq t_{gpu}$, and/or $[N \times$

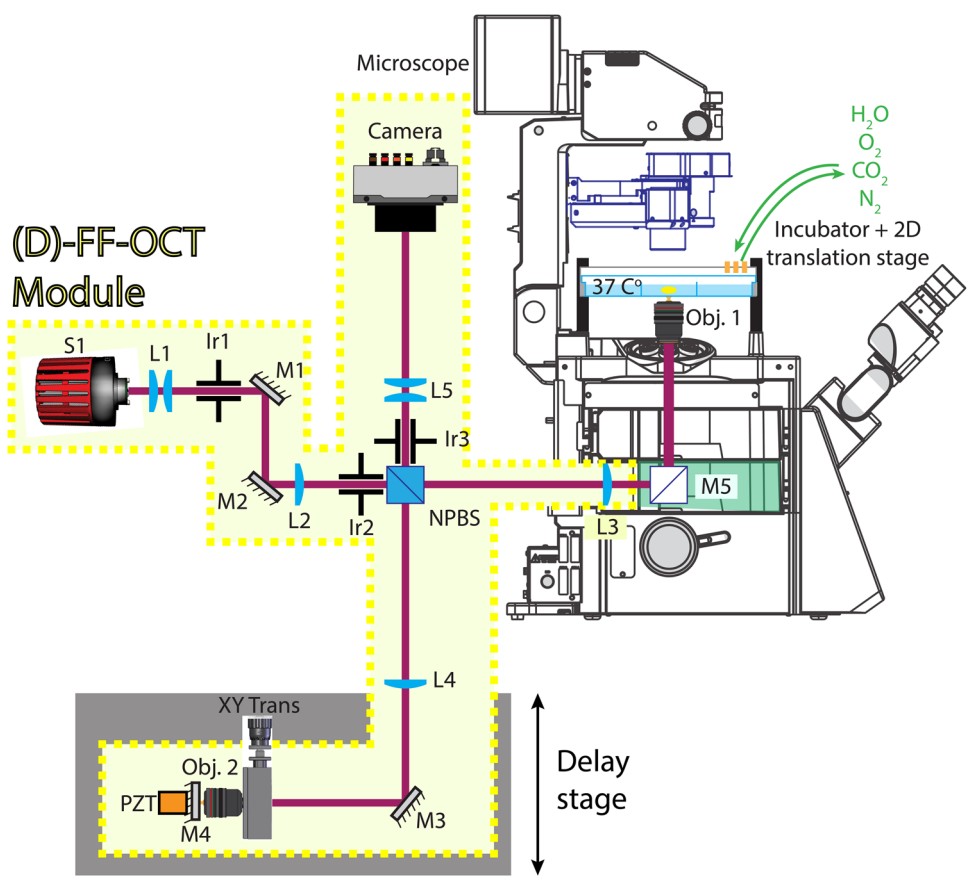

**Fig. 10 Schematic of the set-up.** A mounted light-emitting diode (LED) (M810L3 or M730L5, Thorlabs, Newport, NJ, USA) with a wavelength of 810 nm and 25 nm bandwidth, or 730 nm and 40 nm bandwidth, is used as an extended source (S1) of 1 mm$^2$ with 61.8 $\mu$W mm$^{-2}$, or 13.1 $\mu$W mm$^{-2}$ irradiance, respectively. A first pair of air doublet lenses (L1) is used (AC254-030-B-ML and AC254-150-B-ML, Thorlabs, Newport, NJ, USA) to image S1 onto a first diaphragm (Ir1). A pair of silver mirrors (M1 and M2) (PF10-03-P01, Thorlabs, Newport, NJ, USA) are used to steer the light into the microscope. An air doublet (L2) (AC254-100-B-ML, Thorlabs, Newport, NJ, USA) is used to image Ir1 onto the back focal plane of both objectives (Obj.1 and Obj.2) (UPLSAPO30XSIR, Olympus, Japan) by the intermediate of an air doublet (L3 and L4) (AC254-150-B-ML, Thorlabs, Newport, NJ, USA), respectively. A second iris (Ir2) is imaged onto the sample and M4, a silicon mirror (monocrystalline silicon wafer, Nanomaterials Development Experts Store, China) and is conjugated to a third iris (Ir3) by the intermediary of a non-polarizer-beam-splitter (NPBS), itself conjugated to a cMOS camera (Q-2HFW, Adimec, Netherland) by the intermediary of a pair of air doublet lens es(L5) (AC254-060-B-ML and AC254-200-B-ML, Thorlabs, Newport, NJ, USA). A steering mirror (M3) (PFE10-P01, Thorlabs, Newport, NJ, USA) and a manual translation stage (XY Trans) are used for alignment and fine positioning of Obj.2; standing vertically. A piezoelectric (PZT) (PK25LA2P2, Thorlabs, Newport, NJ, USA) and linear stage (Delay stage) (X-LSQ075A, Zaber, Canada) are used to introduce a fast or fixed phase shift between interferometric fields, respectively. A flat mirror (M5) (CCM1-P01/M, Thorlabs, Newport, NJ, USA) is mounted in the turret (IX3-RFACA-1–3, Rapp Optoelectronic, Germany) of a commercial microscope (IX83, Olympus, Japan). A commercial stage top incubator (H201-K-FRAME, H201-MW-HOLDER, and OBJ-COLLAR-2532, Okolab, Italy) enables temperature and humidity control, and $CO_2$, $O_2$, and $N_2$ gas composition whilst hosting conventional glass-bottomed multiwell plates. A 2-dimensional translation stage (SCANplus IM 120 × 80, 2 mm, Marzhauser, Germany) enables scanning from well to well, as well as mosaicking, and the objective holder of the microscope, mounted on a linear stage (U-MCZ-1-2, Olympus, Japan) enables depth (Z) scanning.

$T_s] + t_p + t_{gpu} \geq t_{save}$—making the acquisition, post-processing and saving limited to $N_{batches} \times [N \times T_s]$. Whilst non-quantitative D-FFOCT has been demonstrated using Holovibes (http://holovibes.com/)[14,84], an acquisition program written in native C++ and CUDA, for fast acquisition and data processing, resulting in a calculation of a dynamic metric every 160 ms, we note that this method used forecasting which works only at a fixed location in the sample and is incompatible with mosaicking and ZStacking. As a result, our acquisition framework is optimal.

This data framework was accomplished with MATLAB, using the toolbox Imaqtool. A function is triggered every $N$ frames, transferring these $N$ frames using getdata on a second RAM CPU thread, whilst the first thread carries on the logging of another batch simultaneously. The data are then transferred to a GPU (GeForce RTX 3090, NVIDIA) where the dynamic metrics are calculated in $t_{gpu} = 1.35$ s for $N = 512$, approximately 7 times

faster than the previous state-of-the-art[68]. The post-processing of the data was rewritten to achieve a speed gain of 3.3. Both scripts were compared on the same GPU with 8 GB of memory for fairer script comparison (GeForce RTX 2070 with Max-Q Design, NVIDIA), resulting in 3.45 versus 12.35 s. Saving of the dynamic metrics on an HDD (4TO Blue S-ATA III 64Mo, WD) is parallelized on another RAM CPU thread using the function parfeval. Overall, the acquisition, data processing, and saving of the dynamic metrics now takes 5.12 s compared to 50.5 s in the previous state-of-the-art[49]. A slower acquisition time of 7.42 s was achieved when mosaicking on free-floating organoids due to the slow translation steps of 2 $\mu$m per 20 ms, as the whole data flow had to be stopped and reinitialized during the slow drift of the 2D translation stage.

The microscope functionalities, as well as the 2D XY translation stage, are interfaced through JAVA functions

developed by μManager[85]—in which MATLAB is coded—and the other instruments are controlled through MATLAB. A dedicated graphic user interface (GUI) was created from scratch for this setup.

Mosaics were reconstructed from tiles with 50% overlapping using MIST (Microscopy Image Stitching Tool)[86], a plug-in available on Fiji[87]. For the free-floating organoids, stage acceleration was kept minimal by micro-stepping with increments of 2 μm in order to avoid sample drifting.

Raw data represents 3GB per stack, or <1 GB once processed for storage.

**Human subject sample preparation**. Postmortem eye tissues used to generate the human iPSC-5F[80] clones were collected in accordance with the French bioethics law at the Laboratory of Anatomy of the Faculty of Medicine of St-Etienne, France. Handling of donor tissues adhered to the tenets of the Declaration of Helsinki of 1975 and its 1983 revision in protecting donor confidentiality. Skin biopsies used to generate the human iPSC-2 clone 7,17[76] were obtained from informed patients under the approval of French regulatory agencies.

**Human iPSC maintenance and retinal differentiation**. Human iPSC 5 F was cultured on truncated recombinant human vitronectin-coated dishes and using the mTeSRTM1 medium (StemCellTM Technologies) as previously described[77,78]. Retinal organoid generation was based on our previously established adherent human iPSC differentiation protocol[76]. Human iPSCs were expanded to 70–80% confluence in 6 cm diameter dishes as described above. At this time, defined as day 0 (d0), human iPSCs were cultured in a chemically defined Essential 6 (E6) medium with 10 units/ml penicillin and 10 mg/ml streptomycin (Thermo Fisher Scientific). At d2, cells were switched to E6N2 medium, composed of E6 medium, 1% N2 supplement, 10 units/ml penicillin, and 10 mg/ml streptomycin (Thermo Fisher Scientific). The media was changed every 2 to 3 days. After four weeks, at d28, identified self-formed retinal organoids were isolated using a needle and cultured as floating structures in ultra-low attachment 24-well plates (Corning) as floating structures in the ProB27 medium supplemented with 10 ng/ml of animal-free recombinant human FGF2 (Peprotech, 100-18B) and half of the medium was changed every 2–3 days[77,78,80]. The ProB27 medium is composed of chemically defined DMEM: Nutrient Mixture F-12 (DMEM/F12, 1:1, L-Glutamine), 1% MEM non-essential amino acids (Thermo Fisher Scientific), 2% B27 supplement (Thermo Fisher Scientific), 10 units/ml penicillin and 10 μg/ml streptomycin. At d35, retinal organoids were cultured in the absence of FGF2 in the ProB27 medium with 10% FBS (Thermo Fisher Scientific) and 2 mM of Glutamax (Thermo Fisher Scientific) for the next several weeks. Around d84, the retinal organoids were cultured in the ProB27 medium with 2% B27 supplement without vitamin A (Thermo Fisher Scientific) until d250[78,81].

**TUNEL and immunostaining experiment**. Sections of retinal organoids were fixed with 4% PAF in PBS for 5 min. TUNEL assays were performed using the in-situ cell death detection kit, TMR red (Roche, Sigma-Aldrich), according to the manufacturer's recommendations. After washes with PBS, nonspecific binding sites were blocked for 1 h at room temperature with a PBS containing 0.2% gelatin and 0.25% Triton X-100 (blocking buffer) and then overnight at 4 °C with the primary antibody (VSX2, 1/200, Santa Cruz SC365519; PAX6, 1/2000, Millipore AB2237 and BRN3A, 1/250, Millipore MAB1585) diluted in blocking buffer. Slides were washed three times in PBS with Tween 0.1% and then incubated for 1 h at room temperature with

appropriate secondary antibody conjugated with either Alexa-Fluor 488, 594, or 647 (Interchim) diluted at 1:300 in blocking buffer with 4',6-diamidino-2-phenylindole (DAPI) diluted at 1:1000 to counterstain nuclei. Fluorescent staining signals were captured with an Olympus FV1000 confocal microscope equipped with 405, 488, 543, and 633 nm lasers. Confocal images were acquired using a 1.55 or 0.46 μm step size and corresponded to the projection of 20 to 40 optical sections.

**Cell counting**. Cell population analysis was performed on confocal images of d28, d35, and d42 retinal organoids with staining of apoptotic cells (TUNEL assay, red), retinal progenitors (VSX2+, green) or total cell population (progenitors PAX6+/VSX2+ and the first differentiated retinal cells, PAX6+/VSX2−, white), as shown in Fig. 6 and described above.

A custom cell population analysis pipeline was first developed in MATLAB in order to automate batch processing with the following steps:

a. Z-stacks were acquired and then collapsed into 2D images using a maximum-intensity projection.

b. Images were then binarized. To do so, the histogram of the pixel values was generated for each image, and the two main peaks were identified, one of which corresponds to the noise and the other to the meaningful signal, in order to find a threshold.

c. The threshold is set as the histogram value, which minimizes the overlap between the two aforementioned peaks; after its identification, all the pixels having a value greater than or equal to the threshold are set to one and the others to zero.

d. Once the binary image mask is obtained, only those regions of the image containing groups of pixels of size greater than 25 are retained in order to avoid over-segmentation and miscounting.

i. An estimation of the number of cells contained in the total mask area is obtained by dividing the total mask area by a fixed number of pixels, corresponding to an arbitrary mean surface per cell factor. In the case of ×20 magnification (pixel size = 0.56 μm), the mean surface area per cell is 50 pixels, corresponding to a circle of radius 2.23 μm. In the case of ×40 magnification (pixel size = 0.26 μm), the mean surface area per cell is 200 pixels, corresponding to a circle of radius 2.07 μm. These two factors correspond to the cell size in 2D at its equatorial plane (see Supplementary Fig. 1).

ii. In the case of colocalization, a logical AND operation is performed on the binarized image for the involved channels before performing the thresholding and size discrimination steps described above.

In the case of the ratio of expressions with respect to DAPI, cell size becomes negligible, as only the number of pixels above the threshold is considered.

The whole custom cell population analysis pipeline is implemented in MATLAB. The code can be provided by the authors upon reasonable request.

Our custom-developed method was validated using a different workflow, and results were compared. Using the Trainable WEKA Segmentation[88] on Fiji[87], two classes were found (i.e., cells versus background), generating a probability map that was then thresholded and binarized using the Otsu algorithm. The Analyze Particle tool was then used in order to count the cells. Comparable results were obtained with the two methods, thus confirming the estimation that we present in Fig. 2.

**Statistics and reproducibility**. Statistical analyses in Fig. 6 represent the mean of at least three independent experiments. Data were averaged and are expressed as means ± SDs (standard deviation scores). Statistical analysis was performed using Prism 9 (GraphPad software) with appropriate statistical tests. A Mann–Whitney test was performed for two group comparisons. Values of $p < 0.05$ were considered statistically significant.

For all experiments, $n = 3$ to $n = 4$ independent replicate organoids or explants under each of the three culture conditions (i.e., young organoids (Figs. 1–7), mature organoids (Fig. 8), explants (Fig. 9)) were imaged. For organoids, replicate meant from the same batch, cultured under identical conditions, while for explants, replicate meant pieces taken from a similar peripheral retinal location from multiple eyes. For the longitudinal experiments (Figs. 1–7), $n = 3$ organoids were followed up over time, with one volumetric image acquired daily over 17 days.

**Retinal organoid sample preparation for D-FFOCT imaging.** Early retinal organoids at d28 were embedded in 3% Matrigel (Corning® Matrigel® Basement Membrane Matrix Growth Factor Reduced Phenol Red Free [Corning, 356231]) in 12-well glass-bottomed plates (IBL, 220.210.042). Embedded structures were cultured in ProB27 medium +FGF2 up to d35 followed by 1 week in ProB27 medium. Half of the medium was changed every 2–3 days[77,78]. Mature retinal organoids around d250 were placed in a black, flat-bottomed glass microscopy-compatible 24-well plate (ibidi), and the well was filled with pre-warmed fresh culture medium. The organoids were left for at least 1 h in the dark in the 37 °C/5% $CO_2$ incubator before imaging.

**Retinal explant.** Porcine eyes were obtained from a local slaughterhouse in agreement with the local regulatory department and the veterinarians from the French Ministry of Agriculture (agreement FR75105131). Eyes were dissected to isolate retinas in a $CO_2$-independent medium (18045054, Thermo), and pieces from the region behind the optic nerve were cut using sterile biopsy punches of 2 mm. Each retinal explant was then placed on a polycarbonate membrane (140652, Thermo, Waltham, Massachusetts) with photoreceptors turned upwards. The explants were then placed in a glass plate (Cellvis, P12-1.5H-N, IBL) for microscopy, with ganglion cells facing up. The level of the medium was precisely controlled to hydrate the retina without completely immersing it to have optimized oxygenation. These pieces were kept in culture in a $CO_2$ incubator at 37 °C for 3 days in Neurobasal-A medium (10888022, Thermo) containing 2 mM of L-glutamine (G3126, MERCK).

**Reporting Summary.** Further information on research design is available in the Nature Portfolio Reporting Summary linked to this article.

## Data availability
Source data for the graphs in the main figures is available as Supplementary Data, and any remaining information can be obtained from the corresponding author upon reasonable request.

## Code availability
Code can be made available upon reasonable request from the authors.

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

## Acknowledgements

We thank Anis Aggoun for suggesting using the MIST plugin from ImageJ. The authors wish to acknowledge funding from IHU FOReSIGHT [ANR-18-IAHU-0001], OPTORETINA (European Research Council (ERC) (#101001841)), Région Ile-de-France Sésame, "OREO" [ANR-19-CE19-0023].

## Author contributions

The overall project was conceived and supervised by K.G. Optical design and construction was conceived and carried out by T.M. supervised by O.T. T.M. conceived and carried out the interfacing, as well as the design and implementation of software architecture and post-processing optimizations (acquisition GUI processing). The method for increasing imaging depth and mosaicking with D-FFOCT was conceived and carried out by TM. Acquisition protocols were designed by S.R., V.F., and T.M., and acquisitions were carried out by T.M., with assistance from SA, on samples provided by S.R., O.G., A.S., S.P., and V.P., with assistance from J.B. and M.C. Images and volumes were reconstructed by T.M. K.G., T.M., O.T., S.R., O.G., and S.A. discussed the results and wrote the article.

## Competing interests

The authors declare the following competing interests: T.M., S.A., S.R., O.T., and K.G. are inventors on a patent application filed by Sorbonne University (number PCT/FR2022/000042, filed 22 April 2022) which covers the modular DFFOCT configuration.
