## [Peer Review File · Communications Biology]

Reviewers' comments:

Reviewer #1 (Remarks to the Author):

Manuscript Summary:

Monfort et al. use dynamic full-field optical coherence tomography (DFFOCT) to longitudinally image organoid and explant cell cultures. DFFOCT is a label-free technique that acquires both structural ('static') and functional ('dynamic') information using interference between a reference beam and back-reflections from the sample. Key advantages of the approach are that the contrast is label-free and z resolution is determined by the bandwidth of the light source rather than the numerical aperture of the objective. Thus, OCT methods are especially useful in clinical applications where exogenous labels cannot be applied to the sample and long working distances (low NA) are required.

Major Comment:

The quality of the work is rigorous, but it is unclear if the results presented are of sufficient interest to the audience of *Communications Biology*. The journal website states that it, "seeks to publish... significant advances across the biological sciences...[or] adjacent research fields where the central advance of the study is of interest to biologists." However, the only biological finding presented is that DFFOCT does not induce gross toxicity during longitudinal DFFOCT imaging. This result may only be interesting to clinically oriented researchers requiring a label-free approach who also use retinal cultures. Biological correlates of the DFFOCT 'dynamic' signals were previously published [1].

The most significant advances are technical aspects of the data handling and system integration: 1) multi-threaded data processing to enable more efficient (faster) acquisition, 2) coupling of the light path into a commercial stand, and 3) GUI-based control of the stand and acquisition parameters. As the author's note, these engineering enhancements are very practically useful, yet they are only described in the Methods and Supplement. We wonder if this manuscript is better suited for a journal that focuses on image data processing or system integration, with the biological results being a supplement to illustrate how these technical improvements enable biological applications that were otherwise impractical on the previously published, less integrated system [2], [3].

Minor Comments:

The authors imply that DFFOCT is less toxic than fluorescence methods and gentle enough for longitudinal studies since the organoids continue to develop over time. Continued growth may not be indicative of completely normal growth. The authors should quantitatively compare DFFOCT-imaged cultures versus cultures imaged only by bright field to rule out minor phototoxic effects more convincingly. Measures of toxicity could include the change in organoid volume over time (growth rate) or application of colorimetric stains that report on some aspect of cell health. We can also imagine terminal measurements based on biochemistry or cytometry. Note that the light powers used in this manuscript are comparable to those used during fluorescence imaging (5-30 W/cm²).

The relevance of the 'dynamic' signals (visually encoded as hue and saturation) is unclear. The morphological descriptions presented would seem to be equally well-supported if the images had been displayed as grey scale, i.e. if the H and B components were omitted. Can the authors show that the dynamic signals contain biologically relevant information that is independent of information in the amplitude signal? Could other label-free methods provide highly similar morphological information, e.g. differential interference contrast microscopy [4]?

The authors previously [2] used a 660 nm light source but now use either 730 or 810 nm. Why were these specific average wavelengths chosen? Is there a theoretical argument for which average wavelength is best for imaging biological samples (all else equal)? Though increased tissue

transparency at longer wavelengths is clearly beneficial for fluorescence imaging, for a reflected light technique it seems that greater transparency also implies less back reflection. Is there a 'happy medium' between these two effects?

References:

1. Groux, K., et al., Dynamic full-field optical coherence tomography allows live imaging of retinal pigment epithelium stress model. *Commun Biol*, 2022. 5(1): p. 575.
2. Scholler, J., et al., Dynamic full-field optical coherence tomography: 3D live-imaging of retinal organoids. *Light Sci Appl*, 2020. 9: p. 140.
3. Scholler, J., et al., Probing dynamic processes in the eye at multiple spatial and temporal scales with multimodal full field OCT. *Biomed Opt Express*, 2019. 10(2): p. 731-746.
4. Lee, J., et al., Cell Membrane Tracking in Living Brain Tissue Using Differential Interference Contrast Microscopy. *IEEE Trans Image Process*, 2018. 27(4): p. 1847-1861.

Reviewer #2 (Remarks to the Author):

Authors Monfort et al. describe a work plan of a D-FFOCT module design to achieve a non-invasive label free 3D longitudinal imaging of retinal organoids. The authors have demonstrated that this module can be coupled to a commercial microscope for ease of use. This is a useful tool for examining 3D in vitro and ex vivo samples. However, are several revisions that are recommended to strengthen the paper and make it accessible to a larger audience.

1. It is not clear why retinal organoids were used and should be explain in the introduction and discussion.
2. Make clear in the introduction what commercial microscope was used as this is mentioned to be key to the new tools accessibility. Eg. BX51 brightfield etc...
3. Make sure to introduce abbreviations in introduction before they appear in results. Eg. Line 75 - Retinal organoid (RO)
4. Line 78 - 80: Please clarify, sentence is not clear
5. Figure 1 - the organisation of the images in the figure are difficult to follow, make the boxes of relatively similar size or the panels approximately the same height.
6. Z=35um move this to figure legends.
7. Scale bar - either put the scale bar on the image or in the figure legend. Please keep it consistent throughout all figures.
8. Figure 2 - more detail in figure legend on what is being shown. Eg. the replication/division of the organoid across time. This would help reader.
9. Figure 4 - not sure about what image c is showing from the figure legend. Also, reconsider using a) and b) again in image C.
10. Methods - dataset size is > 1 To, how large would the datasets be stored? is this a limitation? Address briefly in discussion.
11. In the discussion, a greater explanation of the technical limitations to acquire and process images, even the storage of data would be useful to get a realistic insight into what the incorporation of this tool into the laboratory would realistically entail as it is mentioned that this application is meant to be an accessible modality.
12. Please check general formatting and grammatical errors - Eg. Line 85, (d28) should be (d28). Extra space. Line 119 - extra space between indicate and cellular.

Point by point response to reviewers' comments:

Reviewer #1 (Remarks to the Author):

Manuscript Summary:

Monfort et al. use dynamic full-field optical coherence tomography (DFFOCT) to longitudinally image organoid and explant cell cultures. DFFOCT is a label-free technique that acquires both structural ('static') and functional ('dynamic') information using interference between a reference beam and back-reflections from the sample. Key advantages of the approach are that the contrast is label-free and z resolution is determined by the bandwidth of the light source rather than the numerical aperture of the objective. Thus, OCT methods are especially useful in clinical applications where exogenous labels cannot be applied to the sample and long working distances (low NA) are required.

Major Comment:

The quality of the work is rigorous, but it is unclear if the results presented are of sufficient interest to the audience of Communications Biology. The journal website states that it, "seeks to publish... significant advances across the biological sciences...[or] adjacent research fields where the central advance of the study is of interest to biologists." However, the only biological finding presented is that DFFOCT does not induce gross toxicity during longitudinal DFFOCT imaging. This result may only be interesting to clinically oriented researchers requiring a label-free approach who also use retinal cultures. Biological correlates of the DFFOCT 'dynamic' signals were previously published [1].

The most significant advances are technical aspects of the data handling and system integration: 1) multi-threaded data processing to enable more efficient (faster) acquisition, 2) coupling of the light path into a commercial stand, and 3) GUI-based control of the stand and acquisition parameters. As the author's note, these engineering enhancements are very practically useful, yet they are only described in the Methods and Supplement. We wonder if this manuscript is better suited for a journal that focuses on image data processing or system integration, with the biological results being a supplement to illustrate how these technical improvements enable biological applications that were otherwise impractical on the previously published, less integrated system [2], [3].

We thank the reviewer for these thoughtful comments. We respond firstly as to our choice of journal: this article was written in response to an invitation from the editors for a contribution to a feature issue of the journal, named Collection: Live Microscopy. As stated on the journal website, “with this Collection, we are aiming to build a series of articles that will highlight the current research being done in this area. We will welcome any submissions that fall within the scope of using, extending or improving live microscopy – from new tools to emerging techniques, from conventional to advanced light microscopy.”

We therefore intentionally wished to focus more on technical aspects, to fit with the feature issue theme, than we did in a previous publication in this same journal which sought to correlate DFFOCT signals with biological structures (Groux et al 2022) as the reviewer notes. We have extensively reworked the biological and technical messages of the paper to respond to the comment to improve clarity.

From a biological point of view, the lack of toxicity is one aspect, which we have further explored as described in answer to the minor comment below; but the important biological findings in the organoids which were perhaps insufficiently highlighted are that i) rosette formation begins at the periphery before invaginating into the interior of the organoid. This was never shown before in live imaging and was not detected with conventional imaging methods. ii) mitotic proliferation was shown via live imaging in the organoid interior, which has not been shown before. These points are now highlighted in the abstract, results and discussion sections as follows:

Abstract: “Long term volumetric imaging on human induced pluripotent stem cell-derived retinal organoids is demonstrated, highlighting tissue and cell organization **processes such as rosette formation and mitosis** as well as cell shape **and** motility.”

Introduction, lines 80-82: “The retinal organoid was still alive at the end of this period of time, **and processes such as rosette formation and mitotic proliferation were therefore followed live.**”

Results: lines 146-225 with figures 2-5 (as before)

Discussion: lines 396-402 “Biological results shown include clear 3D views of cone and rod photoreceptors in retinal explants with unprecedented resolution for an OCT-based technique (Fig. 9); two types of iPSC-derived retinal organoids with cell layering distinguishable from the dynamics and morphology of the D-FFOCT signal ; and long term

acquisitions on single organoids over >40 day periods, **highlighting the live follow-up of normally inaccessible dynamic processes such as rosette formation (Fig 3, 5) and mitotic proliferation (Fig 3, 5)**, with the ability to probe mature organoids at later stages of development enabled by the wide field mosaicking and deeper penetration depths.”

In addition, to improve clarity of the biology message, the entire manuscript underwent minor text edits by the biologist authors, and some further biological detail was added in the legend of fig. 2, in material and methods lines 507-590, and in two new figures (6&7) related to the phototoxicity issue raised in the point below which were added.

From a technical point of view, we agree with the reviewer that the main advances that now allow use of the technology by non-specialists including biologists are i) the data processing ii) the integration in module form of the DFFOCT technology into a commercial microscope housing iii) GUI based automated piloting of the acquisitions, but we would add the increased penetration depth, and axial and lateral field size, and also the use of an incubator. There are two novel aspects to this incubator use: firstly, this is the first time we reveal our choice of a stage-top incubator as the best format for our optical design, rather than enclosing the full setup in a controlled environment for example; and secondly, using this incubator meant that this is the first time D-FFOCT could be reproducibly achieved at 37°C and 5% CO₂, enabling imaging under physiological conditions. As noted by the reviewer, these aspects were previously described in the abstract, as well as Methods and Supplement sections.

To improve the clarity of the message, we now explicitly name these aspects “technical results” in the discussion sections beginning line 387:

“here, **our technical results** extend the potential of this promising technique by adapting it to use by a wider community of biologists and other non-optics experts. This was achieved by integrating D-FFOCT to a conventional microscope setup, familiar to biologists, thereby facilitating adoption; allowing continuous protection of the samples under controlled environmental conditions **in a stage-top incubator** suitable for culture; and by precisely automating key aspects of acquisitions including repeatable xyz positioning, 3D stack capture, and mosaicking; along with developing an efficient and fast workflow for acquiring, post-processing and saving datasets. In addition to these protocol improvements, in terms of optical innovation, we showed greater penetration depths achievable with longer wavelength illumination and the improved SNR gained by binning, which opens up the perspective of making faster acquisitions with fewer frames required to calculate dynamic metrics than in past work.”

The data processing aspect has also now been further detailed in results (lines 104-105), discussion (line 374) and material and methods (line 503) sections: **“Raw data represents 3GB per stack, or <1GB once processed for storage.”**

Minor Comments:

The authors imply that DFFOCT is less toxic than fluorescence methods and gentle enough for longitudinal studies since the organoids continue to develop over time. Continued growth may not be indicative of completely normal growth. The authors should quantitatively compare DFFOCT-imaged cultures versus cultures imaged only by bright field to rule out minor phototoxic effects more convincingly. Measures of toxicity could include the change in organoid volume over time (growth rate) or application of colorimetric stains that report on some aspect of cell health. We can also imagine terminal measurements based on biochemistry or cytometry. Note that the light powers used in this manuscript are comparable to those used during fluorescence imaging (5-30 W/cm²).

The authors thank the reviewer for this interesting suggestion.

Firstly, regarding power, while the light power is comparable, the nature of the source and its wavelength is not: we are imaging with a continuous wave source in near infrared wavelengths, while fluorescence imaging uses pulsed sources in the visible range where the photodamage can be considerable in comparison to the infrared [67, 84]. We add these references which relate phototoxicity to wavelength in lines 431-433 of the Methods section:

“Note that although the power applied is similar to levels used in single photon fluorescence microscopy, the continuous wave nature of the source and longer near-infrared wavelengths used here significantly reduce danger of phototoxicity [67,84].”

To demonstrate the lack of toxicity, we add measurements on both illuminated and non illuminated organoids, described in results lines 239-249 with two new figures, 6 and 7, added, a line in discussion 402-404, and the description of the corresponding material and methods in lines 536-590.

Results 239-249: **“To confirm the integrity of retinal organoids during longitudinal acquisition, we performed detection and quantification of apoptosis at single cell**

level, based on labeling of DNA strand breaks (TUNEL experiments, Fig. 6) and immunostaining triggering retinal cell population. In situ cell death analysis coupled with immunostaining confirmed that illuminated d28, d35 and d42 retinal organoids did not present more apoptotic cells compared to those that were not illuminated (Fig. 6a, c, e). At d28, d35 and d42, TUNEL+ cells within non-illuminated retinal organoids represent $2.69\% \pm 0.62$, $1.74\% \pm 0.83$ and $2.43\% \pm 0.64$ and within illuminated retinal organoids $4.17\% \pm 0.19$, $1.81\% \pm 1.2$ and $2.63\% \pm 2.03$ respectively (Fig. 6b, d, f, Supplementary data 2). The number of retinal progenitors (VSX2+/PAX6+ cells) as well as the first differentiated retinal cells (PAX6+/VSX2-) were not impacted. In addition, we confirmed that during image acquisition, the light used for D-FFOCT did not impact the differentiation commitment of progenitors to the retinal ganglion cell (RGC) population (BRN3A/PAX6+ cells, Fig. 7).”

Figure 6: Cell population analysis within non-illuminated or illuminated D28, D35 and D42 retinal organoids. a. Staining of apoptotic cells (TUNEL assay, red), retinal progenitors (VSX2+, green) or total cell population (progenitors PAX6+/VSX2+ and the first differentiated retinal cells, PAX6+/VSX2-, white) in D28 retinal organoids. **b.** Quantification of TUNEL, VSX2 and PAX6 positive cells in D28 retinal organoids. **c.** Staining of apoptotic cells (TUNEL assay, red), retinal progenitors (VSX2+, green) or total cell population (progenitors PAX6+/VSX2+ and the first differentiated retinal cells, PAX6+/VSX2-, white) in D35 retinal organoids. **d.** Quantification of TUNEL, VSX2 and PAX6 positive cells in D35 retinal organoids. **e.** Staining of apoptotic cells (TUNEL assay, red), retinal progenitors (VSX2+, green) or total cell population (progenitors PAX6+/VSX2+ and the first differentiated retinal cells, PAX6+/VSX2-, white) in D42 retinal organoids. **f.** Quantification of TUNEL, VSX2 and PAX6 positive cells in D42 retinal organoids. Data are presented as mean \pm SD (n = 3). Mann-Whitney comparison test. Nuclei were counterstained with DAPI (blue). Scale bar, 50 μ m.

Figure 7: Retinal ganglion cell (RGC) population analysis within non-illuminated or illuminated D42 retinal organoids. Immunostaining of RGCs (BRN3A+/PAX6) in retinal organoids at D42. Nuclei were counterstained with DAPI (blue). Scale bar, 50 μ m.

Discussion 402-404: “We validated that D-FFOCT imaging does not interfere with development of the organoids or induce cell death during longitudinal acquisition (Figs. 6, 7).”

Material and methods 536-590:

“TUNEL and immunostaining experiment

Sections of retinal organoids were fixed with 4% PAF in PBS for 5 min. TUNEL assays were performed using the in-situ cell death detection kit, TMR red (Roche, Sigma-Aldrich) according to the manufacturer’s recommendations. After washes with PBS, nonspecific binding sites were blocked for 1 hour at room temperature with a PBS containing 0.2% gelatin and 0.25% Triton X-100 (blocking buffer) and then overnight at 4°C with the primary antibody (VSX2, 1/200, Santa Cruz SC365519; PAX6, 1/2000, Millipore AB2237 and BRN3A, 1/250, Millipore MAB1585) diluted in blocking buffer. Slides were washed three times in PBS with Tween 0.1% and then incubated for 1 hour

at room temperature with appropriate secondary antibody conjugated with either AlexaFluor 488, 594 or 647 (Interchim) diluted at 1:300 in blocking buffer with 4',6-diamidino-2-phenylindole (DAPI) diluted at 1:1000 to counterstain nuclei. Fluorescent staining signals were captured with an Olympus FV1000 confocal microscope equipped with 405, 488, 543 and 633 nm lasers. Confocal images were acquired using a 1.55 or 0.46 μm step size and corresponded to the projection of 20 to 40 optical sections.

Cell counting

Cell population analysis was performed on confocal images of d28, d35 and d42 retinal organoids with staining of apoptotic cells (TUNEL assay, red), retinal progenitors (VSX2+, green) or total cell population (progenitors PAX6+/VSX2+ and the first differentiated retinal cells, PAX6+/VSX2-, white), as shown in Fig. 6 and described above.

A custom cell population analysis pipeline was first developed in MATLAB in order to automate batch processing, with the following steps:

i.

a. Z-stacks were acquired and then collapsed into 2D images using a maximum intensity projection.

b. Images were then binarized. To do so, the histogram of the pixel values was generated for each image and the two main peaks were identified, one of which corresponds to the noise and the other to the meaningful signal, in order to find a threshold.

c. The threshold is set as the histogram value which minimizes the overlap between the two aforementioned peaks; after its identification, all the pixels having a value greater than or equal to the threshold are set to one and the others to zero.

d. Once the binary image mask is obtained, only those regions of the image containing groups of pixels of size greater than 25 are retained in order to avoid over-segmentation and miscounting.

ii. An estimation of the number of cells contained in the total mask area is obtained by dividing the total mask area by a fixed number of pixels, corresponding to an arbitrary mean surface per cell factor. In case of x20 magnification (pixel size = 0.56 μm), the mean surface area per cell is 50 pixels, corresponding to a circle of radius 2.23 μm . In case of x40 magnification (pixel size = 0.26 μm), the mean surface area per cell is 200 pixels, corresponding to a circle of radius 2.07 μm . These two factors correspond to the cell size in 2D at its equatorial plane (see Supplementary Fig. 1).

iii. In the case of colocalization, a logical AND operation is performed on the binarized image for the involved channels before performing the thresholding and size discrimination steps described above.

In the case of the ratio of expressions with respect to DAPI, cell size becomes negligible, as only the number of pixels above the threshold is considered. The whole custom cell population analysis pipeline is implemented in MATLAB. The code can be provided by the authors upon reasonable request.

Our custom-developed method was validated using a different workflow, and results were compared. Using the Trainable WEKA Segmentation [89] on Fiji [90], two classes were found (i.e. cells versus background), generating a probability map that was then thresholded and binarized using the Otsu algorithm. The Analyze Particle tool was then used in order to count the cells. Comparable results were obtained with the two methods, thus confirming the estimation that we present in Fig. 2.

Statistical analysis

Statistical analyses represent the mean of at least three independent experiments. Data were averaged and are expressed as means \pm SDs (Standard Deviation scores). Statistical analysis was performed using Prism 9 (GraphPad software) with appropriate statistical tests. A Mann-Whitney test was performed for two group comparisons. Values of $p < 0.05$ were considered statistically significant.”

The relevance of the ‘dynamic’ signals (visually encoded as hue and saturation) is unclear. The morphological descriptions presented would seem to be equally well-supported if the images had been displayed as grey scale, i.e. if the H and B components were omitted.

The HSB color scale was first described in Scholler et al and is briefly resumed here in lines 88-90, “Three dynamic metrics were calculated from a time series of 512 FFOCT images (see methods), and were displayed in a hue-saturation-brightness space (HSB) [50]. Red hue indicates faster activity than blue; saturation indicates activity randomness; and brightness captures the axial amplitude, or strength of the activity.”

Fig 1 of Scholler et al depicts and explains the HSB representation in more detail as follows:

processed independently. **c** An intensity trace is plotted for a pixel inside a living retinal organoid. **Post-processing steps d-f.** Dynamic images are computed in the HSV colour space. **d** Hue is computed with the mean frequency, from blue (low temporal frequencies) to red (high temporal frequencies). **e** Saturation is computed as the inverse of the frequency bandwidth; as a consequence, a signal with a broader bandwidth (e.g., white noise) appears dull, whereas a signal with narrow bandwidth appears vivid. **f** The value is computed as the running standard deviation²³. Bottom row is a D29 retinal organoid. **g** Computation of the mean frequency (hue); **h**, frequency bandwidth (saturation); and **i**, dynamic (value) before **j**, reconstruction. Scale bar: 50 μ m

The mean frequency of the recorded intra-cellular motion codes for the hue (H) channel, which represents the colour in the image. The colour ranges from blue, representing low frequencies, to red, coding for high frequencies. The saturation (S) channel is coded by the inverse of the frequency bandwidth of each voxel. For a broad bandwidth, meaning there is a large range of frequencies, the saturation will be low, creating a greyish appearance. On the contrary, for a sharp bandwidth, where a specific frequency is emphasised, the saturation will be high, creating a vivid colour. Finally, the brightness (B), (named value (V) in Scholler et al 2020) which codes for the intensity in the image, is calculated as the standard deviation over a moving window of 50 images, which are then averaged, to give the final intensity highlighting the intra-cellular motion. Finally, the three channels are combined to create a coloured image, representing the dynamic profile of the imaged sample.

Can the authors show that the dynamic signals contain biologically relevant information that is independent of information in the amplitude signal?

While *morphology* can be evaluated using a greyscale amplitude image as suggested, we lose crucial information about *behavior*. Were we to show only the intensity channel as suggested, we would have only a black and white image showing motion amplitude with no information on frequency of intracellular motion (H) nor its bandwidth (S) which are important pieces of biological information as they indicate whether the motion is characteristic of the particular feature. For example, we show the difference between slow vivid blue axons such as in Fig 4, versus fast vivid red mitotic cells such as in Fig 3b-d, i.e. both have a clear characteristic frequency of motion, but the axons are slow while the mitotic cell is fast (therefore using mainly the Hue, or frequency, channel to differentiate the different behaviors). Another example is that we are able to differentiate between inner and outer nuclear layer cells via their color and vividness in Fig 8 (i.e. although of similar size and shape, ONL cells are dim green while INL cells are vivid yellow, meaning that these behavioral features could be used to classify them).

We add this point to the results section, lines 305-309:

“We note that the cells in Fig. 8c and d, contained in different retinal layers, are of similar morphology, but are nevertheless differentiable thanks to the behavioral information rendered in the H and S channels of the color map: the differing frequencies of subcellular motion within these cells show one type to be moving at a faster frequency with at a narrower bandwidth, which can act as a label-free indicator of their identity.”

Could other label-free methods provide highly similar morphological information, e.g. differential interference contrast microscopy [4]?

DIC is mainly used for adherent (2D) imaging. Whilst it has some sectioning ability and may be applied to reconstruct volumes, the overall thickness of the sample must be inferior to a few tens of microns – depending on the sample transparency -- as it is a transmission imaging technique. As a result, retinal organoids, which are > 1mm thick cannot be imaged in 3D using DIC with appreciable sectioning ability. In the reference by Lee et al cited by the reviewer [Lee, J., et al., Cell Membrane Tracking in Living Brain Tissue Using Differential Interference Contrast Microscopy. IEEE Trans Image Process, 2018. 27(4): p. 1847-1861], the cells shown are imaged in 2D and are sparse in suspension. In a thick sample such as an organoid, individual cells cannot be distinguished with DIC. See for example Browne AW, Arnesano C, Harutyunyan N, Khuu T, Martinez JC, Pollack HA, Koos DS, Lee TC, Fraser SE, Moats RA, Aparicio JG, Cobrinik D. Structural and Functional Characterization of Human Stem-Cell-Derived Retinal Organoids by Live Imaging. Invest Ophthalmol Vis Sci. 2017 Jul

1;58(9):3311-3318. doi: 10.1167/iovs.16-20796. PMID: 28672397; PMCID: PMC5495152. where it is stated “Phase contrast images of representative organoids at 46, 89, and 151 days in culture (DIC) demonstrated organoid morphology with limited microstructural detail”. The other important difference is that DIC also cannot show the *behavioral* information of the subcellular activity displayed in the DFFOCT modality in the H and S channels.

We add a reference to Browne et al [74] cited above as this paper reviews many live imaging techniques including phase imaging methods such as DIC in line 54 of the Introduction:

“Among other types of label-free microscopies [74], full-field optical coherence tomography (FFOCT) appears particularly suited to live cell imaging of organoids [50] thanks to its high 3D spatial resolution, contrast based on backscattering and phase differences [8, 12, 13], high sensitivity, and high imaging speed [8, 12-14, 74].”

We also add a phrase in the discussion lines 339-344:

“In comparison to other label-free microscopies, D-FFOCT provides imaging in depth (in contrast to differential interference contrast microscopy), no measurable phototoxicity (Fig 6, 7) and fast acquisition time (in contrast to multiphoton microscopies such as third harmonic generation and coherent anti-Raman Stokes microscopies) and crucially, the dynamic aspect of D-FFOCT provides behavioral information which is inaccessible to these other methods, in addition to revealing the 3D structure.”

The authors previously[2] used a 660 nm light source but now use either 730 or 810 nm. Why were these specific average wavelengths chosen? Is there a theoretical argument for which average wavelength is best for imaging biological samples (all else equal)? Though increased tissue transparency at longer wavelengths is clearly beneficial for fluorescence imaging, for a reflected light technique it seems that greater transparency also implies less back reflection. Is there a ‘happy medium’ between these two effects?

The issue of wavelength choice is addressed in Results, lines 290-291, “longer wavelengths and temporal binning of the data lead to higher SNR level in organoids, enabling imaging at depths up to 230 μm ” and Discussion, lines 349-351, “Use of longer wavelength LED illumination and time series binning allowed acquisition at deeper penetration depths than had been previously demonstrated [50];” Moving to longer wavelengths therefore increases penetration depth, and also reduces phototoxicity [67, 84].

Typically, biological tissues are best imaged in the so-called “therapeutic window” between 650nm and 1350nm where the best compromise between resolution (higher at shorter wavelengths) and penetration (higher at longer wavelengths) is found, and where there is the lowest absorption from typical biological tissue contents such as melanin and blood (see figure below taken from Lim HW, Soter NA. 1993. Clinical Photomedicine. New York: Dekker) in relation to scattering.

Our specific choices of near-infrared wavelengths were limited by the choice of LEDs available from the manufacturer (Thorlabs).

References:

1. Groux, K., et al., Dynamic full-field optical coherence tomography allows live imaging of retinal pigment epithelium stress model. *Commun Biol*, 2022. 5(1): p. 575.
2. Scholler, J., et al., Dynamic full-field optical coherence tomography: 3D live-imaging of retinal organoids. *Light Sci Appl*, 2020. 9: p. 140.
3. Scholler, J., et al., Probing dynamic processes in the eye at multiple spatial and temporal scales with multimodal full field OCT. *Biomed Opt Express*, 2019. 10(2): p. 731-746.
4. Lee, J., et al., Cell Membrane Tracking in Living Brain Tissue Using Differential Interference Contrast Microscopy. *IEEE Trans Image Process*, 2018. 27(4): p. 1847-1861.

Reviewer #2 (Remarks to the Author):

Authors Monfort et al. describe a work plan of a D-FFOCT module design to achieve a non-invasive label free 3D longitudinal imaging of retinal organoids. The authors have demonstrated that this module can be coupled to a commercial microscope for ease of use. This is a useful tool for examining 3D in vitro and ex vivo samples. However, are several revisions that are recommended to strengthen the paper and make it accessible to a larger audience.

1. It is not clear why retinal organoids were used and should be explain in the introduction and discussion.

We are not sure to quite understand the remark about why organoids were used, but the reason we image in retinal organoids is because over half of the authors are experts in retinal organoid research and the imaging modality was developed initially in response to the challenges of imaging in organoids, though it can also be applied in other applications. As this article is focused on the technology, we wished to show a variety of applications (cultured organoids and post mortem explants), but nevertheless in a biology context of interest to our group, who are located in the Vision Institute and are principally concerned with developing retinal organoids and new techniques to image them. Also, our previous work has also focused on retinal organoids and samples, and so we wished to continue on the same sample type in order to compare to state-of-the-art in this technical paper. Indeed it is possible to apply D-FFOCT to other sample types, as mentioned in the introduction lines lines 59-62: "As a result, dynamic and static FF-OCT ((D)-FFOCT), have been used for in vitro and ex vivo studies to specifically resolve most cell types in 3D tissues/organisms [8, 9, 14, 7475, 7576], to identify different cell stage states such as senescence and mitosis [10, 26, 28], and to detect subcellular compartments and organelles [28]." The background to the motivations for the development of organoids in general as a biology model is explained in lines 1-32 of the introduction, and the motivation for the development of DFFOCT to meet their imaging needs is explained in lines 32-85. In summary, as stated in line 32, organoids are highly promising for the fields of "disease modelling, drug testing and development, and personalized medicine [4, 5, 49]." We sought to develop DFFOCT to image them because, line 33-34 "Despite recent advances in microscopy, high-resolution imaging of organoids,

and in particular longitudinal, live, volumetric imaging, is still complex and remains an open challenge [50].”

To clarify the motivation for the specific choice of retinal organoids, we add a phrase in the introduction line 76-78 **“Although DFFOCT can be used in other types of sample, we chose here to focus on retinal organoids in order to compare technical performance of the new module with the state-of-the-art [50].”**

We add a phrase beginning line 346 of the discussion to resume these points:

“Organoids are highly promising for the fields of disease modelling, drug development, and gene and cell therapies [4, 5, 49]. To effectively image organoids, we sought to develop a technology that could meet the requirements of high-resolution, long term, live, volumetric imaging, with improved performance and accessibility compared to our previous device [50].”

2. Make clear in the introduction what commercial microscope was used as this is mentioned to be key to the new tools accessibility. Eg. BX51 brightfield etc...

The brand and model of microscope has been added to the introduction, line 70 “commercial microscope (**IX83, Olympus, Japan**).”, though it should be noted that adaptation to other microscope models with an accessible optical port is completely possible and would acquire only a custom design of the interfacing optics (possibly a change of only one lens for example). We add this point in line 72-73, **“The design presented here could easily be adapted to other microscope brands and models with an accessible optical port by choosing appropriate alternate interfacing optics”**.

3. Make sure to introduce abbreviations in introduction before they appear in results. Eg. Line 75 - Retinal organoid (RO)

Done

4. Line 78 - 80: Please clarify, sentence is not clear

The lines referred to are “Finally, we demonstrate a new method for imaging retinal organoids larger than 1 mm², older than 250 days, at a stage when photoreceptor cells are fully differentiated with the presence of outer segments. This represents an increase of the axial range by a factor of 2 and the transverse range by a factor 4 compared to prior state-of-the-art.”

We replace these with **“Finally, we demonstrate a new method for imaging retinal organoids larger than 1 mm², older than 250 days, at a stage when photoreceptor cells are fully differentiated with outer segments present. Our ability to image these large organoids demonstrates the increase of our axial and transverse imaging ranges by factors of 2 and 4 respectively compared to prior state-of-the-art [50].”**

5. Figure 1 - the organisation of the images in the figure are difficult to follow, make the boxes of relatively similar size or the panels approximately the same height.

Done

6. Z=35um move this to figure legends.

Done

7. Scale bar - either put the scale bar on the image or in the figure legend. Please keep it consistent throughout all figures.

Done (moved to legend for all figures)

8. Figure 2 - more detail in figure legend on what is being shown. Eg. the replication/division of the organoid across time. This would help reader.

The legend for this figure has been updated as follows:

“Figure 2: D-FFOCT volumetric and longitudinal imaging of one single organoid at a depth of 50 μm , across 17 days. Hue scales from 3 to 13 Hz mean frequency. **Evolution of cells and structures could be followed up each day at subcellular resolution in the same organoid. The organoid grows daily, evolving from a spherical ball of retinal progenitors at d28, to a non-descript shape with both long and spherical rosettes, forming from d36, which reflects the proliferation of retinal progenitors leading to an increase of neuroepithelial tissue.** Image mosaicking covers 406x406 μm^2 , 2928x2928 pixels, (3x3), at day 28 (d28) to 717x717 μm^2 , 5163x5163 pixels, (6x6), at day 42 (d42). The scale bar represents 50 μm and stands for all panels.”

9. Figure 4 - not sure about what image c is showing from the figure legend. Also, reconsider using a) and b) again in image C.

This image (now figure 3b-d) shows a mitosis at a series of consecutive time points, spaced by 8 minute intervals. It has been grouped with Fig 3a to make it clearer that this figure address time evolving processes over timescales of days (rosette formation) to minutes (mitosis). The legend has been updated as follows:

“Figure 3: D-FFOCT longitudinal time tracking of a rosette formation over eight days (a) **and mitosis of a retinal progenitor cell over 16 minutes (b-d).** Hue scales from 3 to 13 Hz mean frequency. a, A single rosette found in the large volume displayed in Fig.2 was tracked throughout each daily volume imaged. All images share the same scaling, with images from day 28 to 34 covering 70x70 μm^2 , 500x500 pixels, and images from day 35 to 36 covering 105x105 μm^2 , 750x750 pixels. **b** highlights a spherical cell in a prometaphase with two spindle poles. **c** highlights **the same cell, 8 minutes later, which has become elongated**, characteristic of the anaphase. **d** highlights **the two resulting** daughter cells, **8 minutes later again**, after the telophase. Images cover 40x40 μm^2 , 290x290 pixels. 8 minutes separate each image. Scale bar, 20 μm .”

10. Methods - dataset size is > 1 To, how large would the datasets be stored? is this a limitation? Address briefly in discussion.

Our apologies for this >1To estimation which is misleading. This represents all data from the entire study. Each individual image stack is a far more modest size, e.g. one z stack of raw data is 3GB. The longitudinal study of Fig. 2 for example represents 100GB of raw data in total. These volumes are not limiting aside from storage cost. However, we can delete raw

data once processed to reduce stored volume, so that 3GB per stack raw data is reduced to 1GB per stack of processed data. Individual image slices from each stack are smaller again. We add this information about data storage volume in results (lines 105-106) “**Each raw data stack is 3GB, while processed data can be reduced to <1GB per stack for storage.**”, discussion (line 375) “**data storage is reduced to <1GB per stack after processing**” and material and methods (line 504) sections: “**Raw data represents 3GB per stack, or <1GB once processed for storage.**”

11. In the discussion, a greater explanation of the technical limitations to acquire and process images, even the storage of data would be useful to get a realistic insight into what the incorporation of this tool into the laboratory would realistically entail as it is mentioned that this application is meant to be an accessible modality.

Information was added to the discussion on acquisition, processing and data storage in lines 373-376: “parallel management of data and optimization of post-processing algorithms enabled close to 10 times faster acquisitions than previously demonstrated [8, 14, 50, 69], which is critical for imaging large samples, volumetric imaging and achieving live imaging, while data storage is reduced to <1GB per stack after processing.”

These issues are addressed at greater length in materials and methods with a dedicated section devoted to optimization of data workflow (acquisition, postprocessing and saving) in lines 466-505, and the workflow is illustrated in supplementary fig. 7.

“Optimization of data workflow (acquisition, postprocessing and saving) Acquisitions are triggered through hardware using a transistor-transistor logic of 0-5 V, generated by an acquisition card (NI cDAQ-9174, National Instruments, TX, USA), and data from the camera (11 bits) are transferred via a 4 CoaxPress cables (25 Gbit/s) to a frame grabber (Cyton CXP4, Bitflow, Massachusetts, USA) which is connected to a PCIe 3.0 (32 Gbit/s) of a PC motherboard (WS X299 SAGE, ASUS, Taiwan). Data from the camera are converted to 16 bits by the frame grabber and are then transferred on a parallel thread of the random-access memory (RAM) (8x Vengeance LPX 1X16GB 3000MHZ, Corsair, California, USA) of the motherboard, and logged onto this same thread. Using a multi-threaded architecture, the effective acquisition time T_{tot} is bottlenecked to the irreducible time needed for data to be logged ($t_{logging}$). For example, for generating 10 D-FFOCT images (H,S,B), from $N_{batches} = 10$ batches, on a locked location, the total time of acquisition is $T_{multithreadtot} = 10 \times [N \times T_s] + t_p + t_{gpu} + t_{save}$ rather than $T_{tot} = 10 \times [N \times T_s + t_p + t_{gpu} + t_{save}]$ as in the previous state of the art [69], where T_s is the inverse of the frame rate of acquisition, N the number of images per batch, t_p the time to transfer the data onto the computing RAM, t_{gpu} the post-

processing time on graphical processing unit (GPU) and t_{save} the time it takes to save the post-processed data. The general work flow is sketched in Supplementary fig. 7. As a result, continuous and lossless acquisitions are possible and the effective generation of dynamic metrics tends ($N_{\text{batches}} \rightarrow \infty$) to the incompressible time it takes to acquire the frames themselves—as long as $[N \times T_s] \geq t_p$, and/or $[N \times T_s] + t_p \geq t_{\text{gpu}}$, and/or $[N \times T_s] + t_p + t_{\text{gpu}} \geq t_{\text{save}}$ —making the acquisition, post-processing and saving limited to $N_{\text{batches}} \times [N \times T_s]$. Whilst non-quantitative D-FFOCT has been demonstrated using Holovibes (<http://holovibes.com/>) [14, 85], an acquisition program written in native C++ and CUDA, for fast acquisition and data processing, resulting in calculation of a dynamic metric every 160 ms, we note that this method used forecasting which works only at a fixed location in the sample and is incompatible with mosaicking and ZStacking. As a result, our acquisition framework is optimal.

This data framework was accomplished with MATLAB, using the toolbox `Imaqtool`. A function is triggered every N frames, transferring these N frames using `getdata` on a second RAM CPU thread, whilst the first thread carries on the logging of another batch simultaneously. The data are then transferred to a GPU (GeForce RTX 3090, NVIDIA) where the dynamic metrics are calculated in $t_{\text{gpu}} = 1.35$ s for $N=512$, approximately 7 times faster than the previous state of the art [69]. The post-processing of the data was re-written to achieve a speed gain of 3.3. Both scripts were compared on the same GPU with 8 GB of memory for fairer script comparison (GeForce RTX 2070 with Max-Q Design, NVIDIA), resulting in 3.45 versus 12.35 seconds. Saving of the dynamic metrics on an HDD (4TO Blue S-ATA III 64Mo, WD) is parallelized on another RAM CPU thread using the function `parfeval`. Overall, the acquisition, data processing and saving of the dynamic metrics now takes 5.12 s compared to 50.5 s in the previous state of the art [50]. A slower acquisition time of 7.42 s was achieved when mosaicking on free floating organoids, due to the slow translation steps of 2 μm per 20 ms, as the whole data flow had to be stopped and reinitialized during the slow drift of the 2D translation stage.

The microscope functionalities as well as the 2D XY translation stage are interfaced through JAVA functions developed by `μ Manager` [86]—in which MATLAB is coded—and the other instruments are controlled through MATLAB. A dedicated graphic user interface (GUI) was created from scratch for this setup.

Mosaics were reconstructed from tiles with 50% overlapping using MIST (Microscopy Image Stitching Tool) [87], a plug-in available on Fiji [88]. For the free-floating organoids, stage acceleration was kept minimal by micro stepping with increment of 2 μm in order to avoid sample drifting.

Raw data represents 3GB per stack, or <1GB once processed for storage.”

Supplement 7

In order to visualize how the acquisitions are carried out in this work, a schematic representation of the data workflow is displayed in Supplementary fig. 7, as described in the methods section “Optimization of data workflow” in the manuscript text.

Supplementary fig. 7: Schematic of the data work flow used for generating D-FFOCT images. A transistor-transistor logic 0-5 V signal was generated from an acquisition card (NI cDAQ-9174, National Instruments, TX, USA) enabling synchronisation of the acquisition with the position of the piezoelectric for FFOCT 2-phases imaging. Period of time (t_E) at 5 V corresponding to the exposure time before capacitor discharge. As a result, images at $1/T_s$ speed were acquired. N frames were acquired per batches before being sent to a parallel thread on the computing process unit (CPU) (CORSAIR 4000D AIRFLOW, AMD Ryzen 5 3600 (3.6 GHz / 4.2 GHz), California, USA) random access memory (RAM) (8x Vengeance LPX 1X16GB 3000MHZ, Corsair, California, USA). Whilst a second batch is being acquired, the first batch, now transferred on the CPU RAM, used as a buffer, is transferred in MATLAB working space on a second CPU RAM thread. A third thread on a graphic unit process (GPU) is used for fast calculation in native flop data type, such as fast-Fourier transform. A fourth CPU RAM thread gathers the post-processed data and save them on a solid-state disk. t_p is the time to transfer the data onto the computing RAM, t_{gpu} the post-processing time on graphical processing unit (GPU) and t_{save} the time it takes to save the post-processed data.

12. Please check general formatting and grammatical errors - Eg. Line 85, (d28) should be (d28). Extra space. Line 119 - extra space between indicate and cellular.

Done

Reviewers' comments:

Reviewer #1 (Remarks to the Author):

The authors have mainly addressed my concerns and I support publication. I remain a bit skeptical about how much new and biologically relevant information the method can provide, relative to existing techniques.

Below are some additional comments on statements in the author's rebuttal:

-To argue that D-FFOCT has inherently less phototoxicity than fluorescence imaging, the authors state: "Note that although the power applied is similar to levels used in single photon fluorescence microscopy, the continuous wave nature of the source and longer near infrared wavelengths used here significantly reduce danger of phototoxicity." However, virtually all fluorescence imaging also uses CW light sources (while pulsed sources are even less toxic at a given SNR). NIR wavelengths are less toxic than visible, but again, fluorescence imaging routinely uses this same part of the spectrum (700-800 nm; e.g. NIR dyes). Thus, neither the CW source nor the wavelength range are distinguishing factors that would clearly make D-FFOCT less toxic than fluorescence. A more likely explanation is the absence of fluorophore per se, as fluorophore excitation followed by free-radical generation is the main driver of toxicity (not light alone).

- To argue that the dynamic signal contains information independent of the amplitude signal, the authors state: "We note that the cells in Fig. 8c and d, contained in different retinal layers, are of similar morphology, but are nevertheless differentiable thanks to the behavioral information rendered in the H and S channels of the color map." I am still not convinced that the dynamic signal contains orthogonal information. Put concretely, I bet a machine learning model could be trained to predict the dynamic signal with high confidence based on the amplitude signal alone.

- To argue against DIC, the authors state: "DIC is mainly used for adherent (2D) imaging. Whilst it has some sectioning ability and may be applied to reconstruct volumes, the overall thickness of the sample must be inferior to a few tens of microns – depending on the sample transparency -- as it is a transmission imaging technique." DIC is routinely used 100s of um into tissue and can be implemented in reflection mode when transmission mode is inaccessible. Certainly, DIC and amplitude D-FFOCT signals are not the same, but DIC just might provide quite similar morphological information, at least over the few hundreds of microns imaged here.

Reviewer #2 (Remarks to the Author):

Thank you for addressing each comment in detail.

-To argue that D-FFOCT has inherently less phototoxicity than fluorescence imaging, the authors state: "Note that although the power applied is similar to levels used in single photon fluorescence microscopy, the continuous wave nature of the source and longer near infrared wavelengths used here significantly reduce danger of phototoxicity." However, virtually all fluorescence imaging also uses CW light sources (while pulsed sources are even less toxic at a given SNR). NIR wavelengths are less toxic than visible, but again, fluorescence imaging routinely uses this same part of the spectrum (700-800 nm; e.g. NIR dyes). Thus, neither the CW source nor the wavelength range are distinguishing factors that would clearly make D-FFOCT less toxic than fluorescence. A more likely explanation is the absence of fluorophore per se, as fluorophore excitation followed by free-radical generation is the main driver of toxicity (not light alone).

The reviewer is right, to some extent, that the free radical generation is likely the main driver of phototoxicity, and is mostly brought by the fluorophores themselves. In the revised version, we emphasized the role of fluorophores and replaced the sentence (line 423):

Note that although the power applied is similar to levels used in single-photon fluorescence microscopy, the continuous-wave nature of the source and longer near-infrared wavelengths used here significantly reduce danger of phototoxicity [67, 84]

By:

Note that although the power applied is similar to levels used in single-photon fluorescence microscopy, the absence of excited fluorophores that create oxidative stress, the continuous-wave nature of the source and longer near-infrared wavelengths used here significantly reduce danger of phototoxicity [67, 84]

In ref [84], the phototoxicity of labeled and unlabeled cells are compared, and while it is increased with fluorophores, it is also increased when high power, visible, and pulsed light are used (refs [67, 84]). For instance, NIR sources have energy below the band gap of endogenous fluorophores. Therefore, one photon absorption is very unlikely to occur and create free radical and/or triplet state -- which are toxic (phototoxicity) -- when using NIR CW sources. 2-photon absorption could significantly occur at NIR wavelength if pulsed light would be used (DOI 10.1002/bies.201700003), therefore CW is an important aspect to mention.

Although pulsed light is theoretically not needed and more phototoxic, many commercial microscopes found in biology labs around us still use pulsed lasers. The reviewer is also right to mention that some NIR dyes can be used, but for some reason (to the best of our knowledge) they seem to be quite toxic, and phototoxic (due to free radicals most likely). This article shows the phototoxicity

of NIR fluorophores (<https://doi.org/10.3389/fphys.2023.1126805>). There are many cases also where using NIR fluorophores are not an option, either because a specific line/antibody is not available, or when multiple labeling is used (especially important if a global view of the sample as obtained naturally with D-FFOCT is required in fluorescence).

In contrast we have proven in the revised manuscript that D-FFOCT using CW NIR sources is actually not toxic at all (with no visible effect at least and no phenotypic disruption).

- To argue that the dynamic signal contains information independent of the amplitude signal, the authors state: “We note that the cells in Fig. 8c and d, contained in different retinal layers, are of similar morphology, but are nevertheless differentiable thanks to the behavioural information rendered in the H and S channels of the color map.” I am still not convinced that the dynamic signal contains orthogonal information. Put concretely, I bet a machine learning model could be trained to predict the dynamic signal with high confidence based on the amplitude signal alone.

In the following sentence:

We note that the cells in Fig. 8c and d, contained in different retinal layers, are of similar morphology, but are nevertheless differentiable thanks to the behavioral information rendered in the H and S channels of the color map: the differing frequencies of subcellular motion within these cells show one type to be moving at a faster frequency with at a narrower bandwidth, which can act as a label-free indicator of their identity

we do not claim that the amplitude and H and S channels are independent. We simply relate that it is easier to differentiate the two cell populations from the H and S channels. However, if we acknowledge that retinal cells are not the best example as the few cell populations are easily separated (both in amplitude, and in frequency -H channel and in morphology), we have stronger evidence from our past work that H, S, B channels are orthogonal. We often see cells with the same amplitude signal (Brightness channel) but different central frequency and frequency bandwidth. As demonstrated in the past by Scholler et al (<https://doi.org/10.1038/s41377-020-00375-8>), Fig1g-i, each component displays some spatially uncorrelated behavior, showing orthogonality properties of the 3 dynamic metrics used to display D-FFOCT results. A change in H and S channel can also be a marker of stress, and /or mitotic state. We have also shown that a change of temperature (<https://doi.org/10.1364/BOE.7.004501>) does not significantly change the amplitude signal, but it does affect the central frequency. It can also help to differentiate between Brownian motion (large frequency bandwidth) from active transport in cells that exhibit both states (e.g. beta cells in the pancreas, immune cells or red blood cells - see doi:10.3390/app7030236). To summarize, we have shown in the past that the 3 metrics are orthogonal.

However, we reckon that in some samples, in particular retinal organoids, cells can often be segmented from amplitude and morphological features only. Still, we are convinced that the H and S channels are useful to find biomarkers related to cell state.

In order to mitigate the above sentence, we have slightly rephrased it, and added references (lines 299-302)

We note that the cells in Fig. 8c and d, contained in different retinal layers, are of similar morphology, but are **more easily** differentiable thanks to the behavioral information rendered in the H and S channels of the color map: the differing frequencies of subcellular motion within these cells show one type to be moving at a faster frequency with at a narrower bandwidth, which can act as **another** label-free indicator of their identity, **as previously shown** [50]

- To argue against DIC, the authors state: “DIC is mainly used for adherent (2D) imaging. Whilst it has some sectioning ability and may be applied to reconstruct volumes, the overall thickness of the sample must be inferior to a few tens of microns – depending on the sample transparency -- as it is a transmission imaging technique.” DIC is routinely used 100s of um into tissue and can be implemented in reflection mode when transmission mode is inaccessible. Certainly, DIC and amplitude D-FFOCT signals are not the same, but DIC just might provide quite similar morphological information, at least over the few hundreds of microns imaged here.

Although the reviewer is right that reflection DIC can be installed and can image in depth in transparent samples or at low resolution, we believe that DIC would fail at imaging organoids at high resolution. Unfortunately, we don't have much experience with DIC ourselves as we don't have one accessible in the lab, and can't find any convincing article on that front. If the reviewer has some precise paper or experience, we would be glad to mitigate our point of view. Our only experience with DIC in depth is during the use of patch clamp recordings (see image - as we could not find precise reference or DOI: 10.1007/BF00374949

Spinal cord

Ventral muscle

the surface of the brain). However, we feel this is mostly applicable to transparent samples such as young depigmented zebrafish or at the surface of thick samples. Related to organoids, we could only find references where DIC is applied at low resolution

(<https://www.nature.com/articles/s41598-018-21201-7>), shallow depth, or young organoids (see image and <https://www.nature.com/articles/s41596-021-00607-0>) , or in sliced organoids (<https://www.sciencedirect.com/science/article/pii/S0006291X19319722#fig3>)

In the provided image, although the organoids are only 2-5 days old – and hence remain transparent, the cells cannot be clearly identified, and many artefacts can be seen (as also described in <https://www.ncbi.nlm.nih.gov/pmc/articles/PMC7734411/>)

In general, we have been told by organoid users that DIC imaging cannot perform high resolution imaging.

It also makes sense to us that DIC is more sensitive to scattering and aberrations than OCT. While multiple scattering and aberrations affect OCT by lowering the accessible full well capacity (generating noise as the square root of the number of collected multiply scattered photons), optical aberrations increase the spot size of the two DIC beams, so that they rapidly overlap and see the same phase on average. Multiply scattered photons should also dilute the probed phase in a linear way.

Finally, DIC imaging only records phase gradients, regardless of cell health or state, so that an apoptotic cell would not be distinguishable with DIC.

To summarize, the tens of microns we mentioned was related to organoids, rather than a very general value. If the reviewer has a reference or experience in using DIC in thick organoids, we would be glad to learn and discover that this is possible, and would be ready to advocate for DIC as an alternative in the discussion. Otherwise, since the sentence mentioned by the reviewer was only part of the cover letter and response to reviewer, we would not modify the main document. We hope we have convinced the reviewer here that DIC is not a viable option for organoid imaging (to the best of our knowledge).

REVIEWERS' COMMENTS:

Reviewer #1 (Remarks to the Author):

The authors have addressed all of my concerns. I support publication.

Reviewer #2 (Remarks to the Author):

NA